# Mini Wind Harvester and a Low Power Three-Phase AC/DC Converter to Power IoT Devices: Analysis, Simulation, Test and Design

**Borja Pozo** [1,*] , **José Ángel Araujo** [2] , **Henrik Zessin** [3] , **Loreto Mateu** [3] , **José Ignacio Garate** [2] and **Peter Spies** [3]

1. Electronics and Communications Unit, Tekniker, Calle Iñaki Goenaga 5, 20600 Eibar, Spain
2. Department of Electronics Technology, University of the Basque Country (UPV/EHU), 48080 Bilbao, Spain; joseangel.araujo@ehu.eus (J.Á.A.); joseignacio.garate@ehu.eus (J.I.G.)
3. Integrated Energy Supplies, Fraunhofer IIS, Nordostpark 84, 90411 Nuremberg, Germany; henrik.zessin@iis.fraunhofer.de (H.Z.); loreto.mateu@iis.fraunhofer.de (L.M.); peter.spies@iis.fraunhofer.de (P.S.)
* Correspondence: borja.pozo@tekniker.es

**Abstract:** Wind energy harvesting is a widespread mature technology employed to collect energy, but it is also suitable, and not yet fully exploited at small scale, for powering low power electronic systems such as Internet of Things (IoT) systems like structural health monitoring, on-line sensors, predictive maintenance, manufacturing processes and surveillance. The present work introduces a three-phase mini wind energy harvester and an Alternate Current/Direct Current (AC/DC) converter. The research analyzes in depth a wind harvester's operation principles in order to extract its characteristic parameters. It also proposes an equivalent electromechanical model of the harvester, and its accuracy has been verified with prototype performance results. Moreover, unlike most of the converters which use two steps for AC/DC signal conditioning—a rectifier stage and a DC/DC regulator—this work proposes a single stage converter to increase the system efficiency and, consequently, improve the energy transfer. Moreover, the most suitable AC/DC converter architecture was chosen and optimized for the best performance taking into account: the target power, efficiency, voltage levels, operation frequency, duty cycle and load required to implement the aforementioned converter.

**Keywords:** energy harvesting/harvester; reactor wind energy; operation mode; three-phase; AC/DC converter

## 1. Introduction

Presently, smart electronic systems and wireless sensor network technologies are widespread in home and industry environments [1–3]. Their influence over people's daily activities such as health and well-being, security, education, entertainment has increased substantially [4]. Moreover, there is taking place a change in automation and control process as smart sensors for communication, control and interoperability are being integrated within enterprise business systems [3]. Furthermore, efficiency and interconnectivity of the Industry [5] are improving with the aid of the aforementioned technologies and systems.

Nowadays, smart electronic systems and wireless sensor networks must be whole in order to achieve low cost and energy efficiency, which is one of the main features [6], especially when aiming for green electronics [7]. Therefore, energy harvesting becomes the most appropriate sources of energy for these electronic systems. Furthermore, energy harvesting systems have become a key technology to gather energy from the environment and give answer to some of the technical challenges that face the

deployment of wireless smart sensor networks [8,9], namely their independence from the mains power grid [10].

An energy harvester is a transducer devoted to obtaining the maximum amount of energy from a given physical phenomenon [11]. Usually, an energy harvesting system has three main elements [12]: a harvester, a low power management system, and a low power storage system. The harvester's characteristics determine the type of power conversion required: Direct Current/Direct Current (DC/DC) or Alternate Current/Direct Current (AC/DC) [13], which sets the specifications of its architecture and conditions the power device selection. Summarizing, there are harvesters that collect energy in DC form and others that provide AC signals, such as wind or liquid flow harvesters. Consequently, the power converter of the system must be suited to the waveforms and energy that the harvesting technology targets. Hence, designing a whole energy harvester system [14,15] requires specific research of the available converter.

The core idea of dynamic fluid harvesters is to use large-scale generator technology principles in small-scale generators [16]. Furthermore, currently three types of generators are used to harvest energy from the wind [17,18]: electrostatic, piezoelectric, and electromagnetic. The electrostatic harvesters base their operational principles on the friction between electrocharged materials, which induces variations of capacitance [19]. This charge is converted into electricity thanks to electrodes [20]. However, these generators are not very efficient, and the power output levels they provide are low compared with other technologies listed in [18]. The piezoelectric (PZT) harvesters base their operation principles on a vibration and deformation of piezoelectric materials [16]. Early PZT harvesters used a mechanical kinetic source to generate energy, but the state of the art PTZ harvesters also use them as the energy source [21]. These harvesters include mechanical parts into the PZT material to optimize and increase its efficiency [22]. The electromagnetic (EM) harvesters base their operational principles on the Faraday and Lenz laws. The rotation of magnets produces an electromagnetic field that is harvested and converted to electrical energy with inductive coils [18]. On the one hand, PZT harvesters produce more energy when the resonance frequency is achieved, but on the other hand, EM harvesters increase the power generation when the rotational or linear velocity is increased on the harvesting device [23,24].

One of the challenges with small electromagnetic generators is that, due to the reduced size, the output voltage level only reaches a few hundreds of millivolts in AC, but the operational electronics need a few volts (i.e., 3.3 V) in DC to work. In the literature there are three main techniques of AC/DC conversion [25]: direct rectification of AC signals into DC, AC rectification plus the DC/DC conversion and direct AC–DC conversion. The first alternative only rectifies the input signal, thus providing a low signal level due to losses in the rectifier. In the second one, the signal is rectified into DC first and it is transformed later into the desired DC level with a DC/DC converter, usually a boost converter that steps up the voltage: if compared with other alternatives, the number of steps required reduces its efficiency. For example, [26] proposes a full-bridge AC/DC rectifier with a DC/DC buck-boost converter. In the first stage, the rectifier has 34% efficiency. The buck-boost converter efficiency of the second stage has power losses between 18% and 46%. Thus, the two-stage system efficiency is reduced to a 54.12% in the best case, basically due to the rectifier losses. Another example of two stage power conversion is shown in [27]. This research uses a full-bridge rectifier and a Synchronous Electrical Charge Extraction (SECE) topology with shared inductor. In this case, three piezoelectric harvesters generate the AC input signal, where a full bridge rectifier is included per each harvester, and the three inputs in a single SECE converter for the DC/DC operation are combined. This solution improves the efficiency of the system; however, a multisource system is needed. Ref. [28] develops an AC/DC direct converter, but the efficiency achieved is only 45%. Their solution consists of a direct AC/DC bridgeless boost converter and a diode voltage multiplier. The voltage multiplier could be an alternative to avoid the bridge rectifier, however, the efficiency of this type of architecture is not the best due to the input voltage levels and the use of diodes in the first stage of the architecture.

One of the challenges of small wind harvesters is the low power that they deliver to the load. Therefore, the aim of this research is to improve the small wind harvester design, achieving

higher power output. Thus, in the first stage of this work, the mini-wind harvester is defined as a three-phase equivalent model, and later, the system is developed, tested and verified with experimental measurements to demonstrate the improvement of the efficiency and power output and show the improvement in the state of the art of current wind harvesters.

The other issue analyzed regarding the current progress of the technology is the power management architectures for the AC generators, specifically to those that generate microvolts, such as the electromagnetics. As has been remarked, the two-stage power manager introduces high losses to the system, basically, due to the diodes' losses at the rectifier stage. The objective of the present work is to develop an efficient three-phase converter translating the efficient converters from power electronics concepts to low power electronics. Thus, eliminating the rectifier stage and the two stage converters. So, this paper analyzes different architectures, suggests devices and evaluates the parameters that characterize the performance of AC/DC converter, aiming for voltage, power, efficiency, and current waveforms. The results obtained are used to identify the most suitable architecture, devices, and parameters in order to obtain the best performance and maximum efficiency of the harvester for the proposed AC/DC converter architecture.

## 2. Harvesting System

The whole AC harvester system comprises the wind flow harvester, the AC/DC converter and the storage device. The aforementioned system converts the wind potential energy into mechanical energy, then into electromagnetic energy and afterwards into electrical energy. In a first stage, the power of wind pressure is harvested and transformed into mechanical power with an efficiency $\eta_1$. Next, the mechanical power is converted into electrical energy through the electromagnetic field generated, with an efficiency of $\eta_2$. Then an AC/DC converter adapts the levels of the harvested AC signal into the required DC, with an efficiency of $\eta_3$. Finally, the energy is stored for later use or it is directly employed to power any given electronics. Figure 1 shows the system block diagram and the energy conversions with their respective efficiencies.

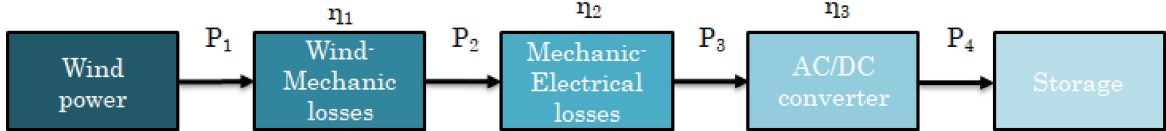

**Figure 1.** Block diagram of complete harvester Alternate Current (AC) system.

Equation (1) provides the harvesters power balance as the product of the system's conversion efficiencies and the input light power:

$$P_4 = \eta_{total} \cdot P_1 = (\eta_1 \cdot \eta_2 \cdot \eta_3) \cdot P_1 \tag{1}$$

## 3. Mini Wind Harvester Electro-Mechanical Three-Phase Model, Simulation, and Comparison with Test Results

### 3.1. Mini Wind Harvester Description

Whenever there is a flow of gas falling through a mini wind harvester, its blades, axis, bearings and two rings which contain magnets start rotating. This movement generates a magnetic field that is harvested and converted into electrical energy through the coils. The system block diagram is shown in Figure 2a and the harvester itself in Figure 2b [29].

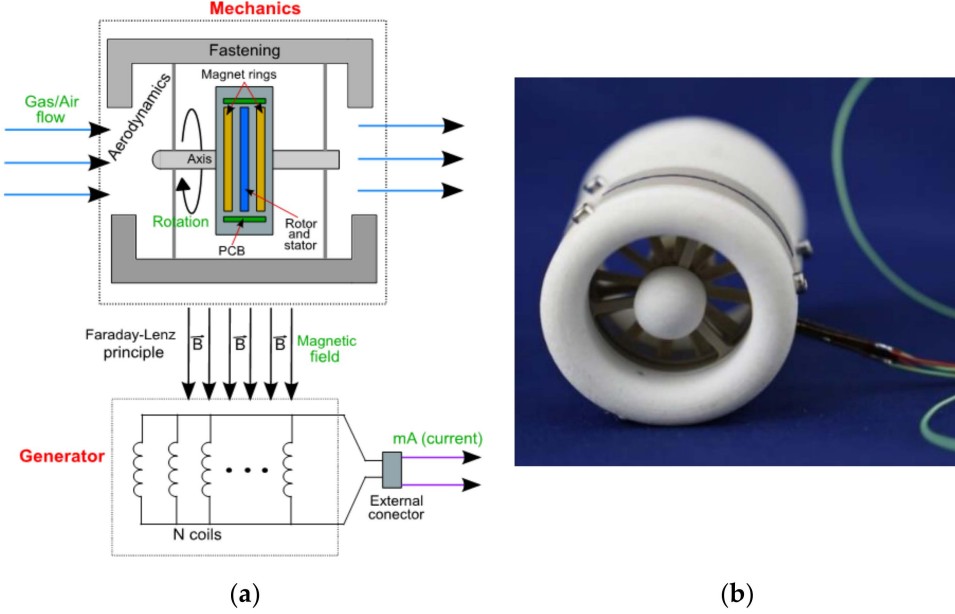

(**a**)                                    (**b**)

**Figure 2.** Used wind energy harvester and its operation diagram. (**a**) Operation diagram. (**b**) Mini wind energy harvester (Inside diameter 22.7 mm, and outer 27.7 mm).

The designed system integrates into a single mechanical body the rotor, blades, and a magnetic ring. Figure 3 shows the details of the magnetic rings: the red and blue colors represent the magnets, the light grey represents the rings and the dark grey represents the coating of nickel. The magnet employs a total of 32 magnets (16 poles) of NdFeB (Neodymium).

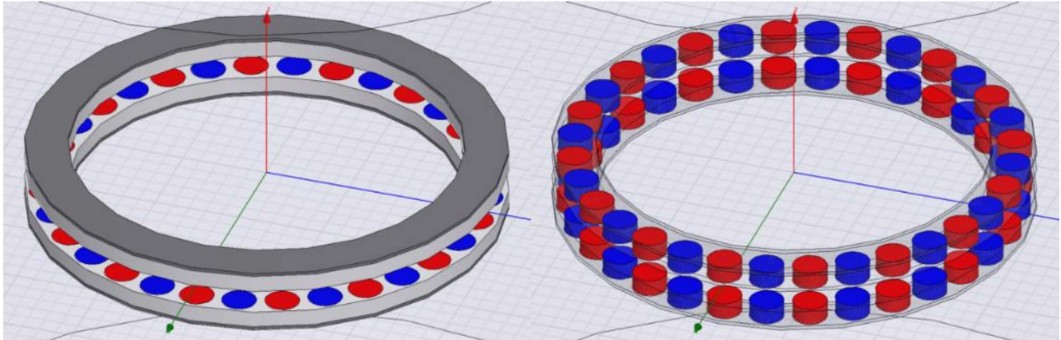

**Figure 3.** Magnet rings details.

The arrangement of the magnets is as follows:

- The red and blue magnets have opposite polarity to generate an alternating magnetic flux when a continuous rotational movement is produced.
- The opposite magnets in two rings have the same polarity in order to not override the magnetic field in the center.

The electrical part is composed of a three-phase generator. The basic Printed Circuit Board (PCB) coil cell is a square coil, Figure 4a, that is cloned n times, Figure 4b. These coils were configured taking into account the number of turns necessary for them to be inserted into the PCB and the size of the magnets. Moreover, so as to increase the available winding room and achieve the highest feasible L, the coils have a square concentric architecture with five turns. The longest turn has a 6.05 mm length and shortest 2.25 mm. In addition, the clearance between traces is set up to 50 μm and the trace weight is 80 μm. The coils are placed on both sides of the PCB. These coils are connected in series, thus enabling a single current path without canceling the total magnetic flux. The total number of double side coils

that fit in this ring is 48. However, each phase is made of 16 inductors. Series connection between both side coils is done through vias located in the center/final coil of the square inductors, as Figure 4a's detail shows.

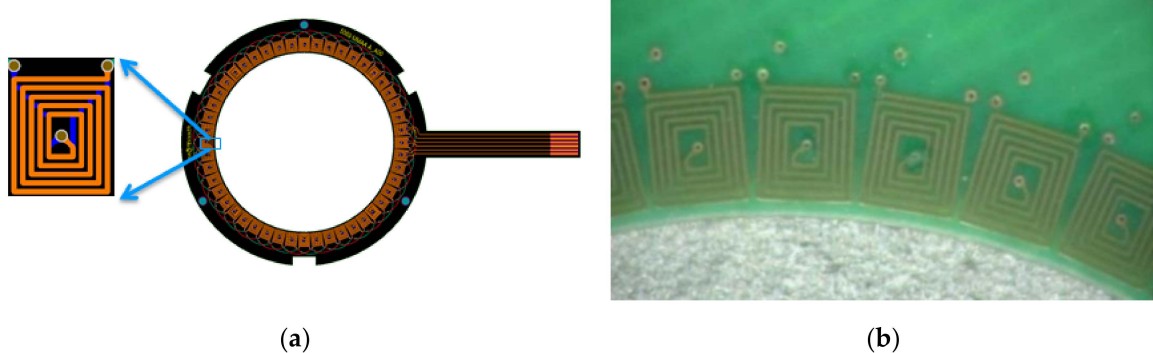

(**a**)          (**b**)

**Figure 4.** Three-phase generator PCB. (**a**) PCB design. (**b**) Manufactured coils and connection drills.

### 3.2. Theoretical Principles of Mini Wind Harvester Operation

Whenever the harvester is in an open space, it harvests the kinetic energy of the air, i.e., wind velocity. However, when the harvester is placed in a pipe, it harvests the potential energy of the air, i.e., wind pressure inside the pipe. Figure 5 shows the described situation.

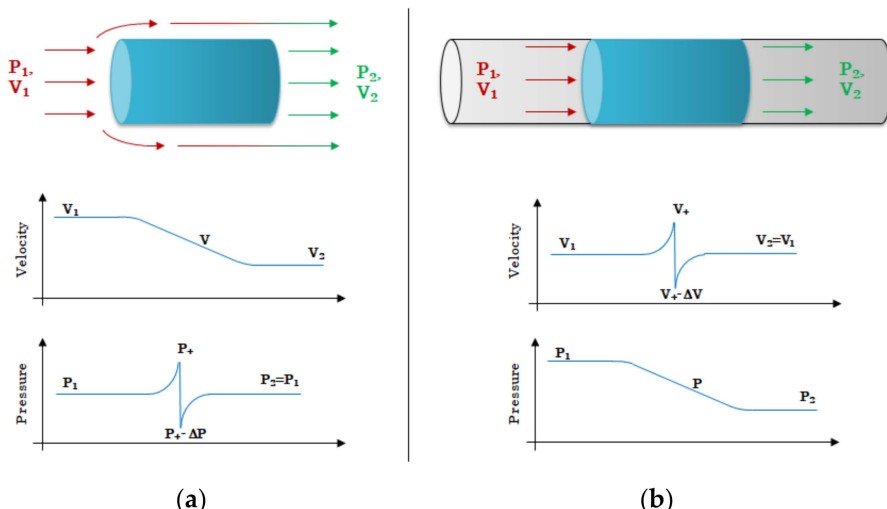

(**a**)          (**b**)

**Figure 5.** Open space operation principle versus test-bench operation principle. (**a**) Harvester in an open space. (**b**) Harvester in a pipe.

In both cases, the law of conservation of energy applies. In Figure 5a, potential energy on one side cancels out the other side and the harvested energy comes from wind kinetic energy. In Figure 5b, the harvester has the same behavior, but the kinetic energy is cancelled, and the harvested energy comes from the wind potential energy. The described behavior is expressed as in Equation (2):

$$E_{potential1} + E_{kinetic1} = E_{potential2} + E_{kinetic2} + E_{harvested} \tag{2}$$

where $E_{potenital1}$ and $E_{potenital2}$ are the potential energy of the wind before and after the harvester, $E_{kinetic1}$ and $E_{kinetic2}$ are the kinetic energy of the wind before and after the harvester, and $E_{harvested}$ is the energy harvested by the wind harvester.

On the one hand, if the turbine is installed in a windy open space, the absolute potential energy is zero, and the harvested power will come only from the kinetic one, as expressed in Equation (3),

the potential energy being equal at both sides ($P_1 = P_2$). In addition, due to the energy principle theorem, $v_1$ must be bigger than $v_2$, $v_1 > v_2$, and consequently $\Delta v = v_1 - v_2$.

$$E_{harvested} = \frac{1}{2}\rho\left(v_1^2 - v_2^2\right) \tag{3}$$

When the turbine starts working under the required threshold wind condition, the rotor will rotate, harvest, and convert the kinetic energy of the wind into mechanical energy. The kinetic power of the wind, $P_{wind}$, using the aerodynamic equation of [30], is given by Equation (4). The transformation of wind power into rotational power by the harvester rotor is a complex aerodynamic phenomenon as described in [31–33]. Ideally, the power extracted from the ambient wind, $P_{aero}$, could be expressed as Equation (4):

$$P_{wind} = \frac{1}{2}\rho v_1^3 A \rightarrow P_{aero} = \frac{1}{2}\rho v_1^3 A C_p(\lambda, \theta) \tag{4}$$

where $\rho$ is air density, $v_1$ is air velocity, A turbine area, and $C_p$ Betz's law coefficient. This power coefficient or the aerodynamic efficiency of the rotor, which is a nonlinear function of the pitch angle $\theta$ of the harvester blades and tip speed ratio $\lambda$. $\theta$ is the angle of the blade element concerning the plane of rotation and $\lambda$ is quotient between the tangential speed of the rotor blade tips and the incoming wind speed, expressed as Equation (5):

$$\lambda = \frac{\omega r}{v} \tag{5}$$

where $\omega$ is the angular velocity (rad/s) and r the radius of the rotor. The power coefficient, $C_p (\lambda, \theta)$ Betz power coefficient equation is shown in Equation (6):

$$C_p(\lambda, \theta) = C_1\left[\left(\frac{c_2}{\lambda_i}\right) - c_3\theta - c_4\theta^{c_5} - c_6\right]e^{\left(-\frac{c_7}{\lambda_i}\right)} \tag{6}$$

where $\lambda_i$ is obtained with Equation (7):

$$\lambda_i = \frac{1}{\left[\frac{1}{(\lambda + c_8\theta)}\right] - \left[\frac{c_9}{(\theta^3 + 1)}\right]} \tag{7}$$

The theoretical values and generated power were calculated to verify if the operation principle matches the results obtained. In addition, some of the dimensional and mechanic characteristics of the micro-reactor were measured and provided in Table 1.

**Table 1.** Micro-reactor dimensional-mechanic characteristics.

| System Values | |
|---|---|
| $\rho$ | 1.225 |
| A_harvester | 0.00045 |
| r_harvester | 0.016 |
| r_rotor | 0.00355 |
| length_blades | 0.01476 |
| $\theta$ | 34.45 |
| weight | 0.0032 |
| length_swept | 0.024 |
| A_small_pipe | $3.31 \times 10^{-5}$ |
| A_big_pipe | 0.000829 |

Refs. [34,35] propose Betz power coefficients for large turbines. However, in works [31,36] with small size wind turbines, the Betz power coefficients are different, see Table 2. Hypothetically, the second coefficients are adequate for this work. However, for this system, both sets of coefficients were verified

with mathematical calculations [37]. The analysis of these results shows that the coefficients chosen for the mini-turbines are more adequate.

**Table 2.** Used Betz power coefficients for mathematical calculation of operation parameters.

| $C_p$ **Coefficient** | $c_1$ | $c_2$ | $c_3$ | $c_4$ | $c_5$ | $c_6$ | $c_7$ | $c_8$ | $c_9$ |
|---|---|---|---|---|---|---|---|---|---|
| CSMWT | 0.6 | 160 | 0.93 | 0 | 0 | 9.3 | 9.8 | 0.037 | 0 |

### 3.2.1. Work Operation at Open Space Environment

The parameter wind flow is obtained by multiplying harvester area with flow velocity. Finally, for $P_{wind}$, $P_{aero}$, $C_p(\lambda,\theta)$, $\lambda_i$ and $\lambda$ parameters, calculation was performed from Equations (4)–(7), respectively. Table 3 and Figure 6 show the achieved results.

**Table 3.** Wind and mechanical calculated values at open space condition.

| $\omega$ (rpm) | $\omega$ (rad/s) | Velocity (m/s) | $P_{wind}$ (mW) | $P_{aero}$ (mW) | $C_p(\lambda,\theta)$ | $\lambda_i$ | $\lambda$ | $Q$ (m³/s) |
|---|---|---|---|---|---|---|---|---|
| 9495 | 997 | 3 | 13.719 | 3.622 | 0.264 | 2.454 | 1.180 | 0.002 |
| 14,271 | 1498 | 4 | 32.518 | 9.104 | 0.280 | 2.605 | 1.330 | 0.003 |
| 19,344 | 2031 | 5 | 63.513 | 18.147 | 0.286 | 2.717 | 1.442 | 0.004 |
| 24,057 | 2526 | 6 | 109.750 | 31.443 | 0.286 | 2.769 | 1.495 | 0.005 |
| 26,180 | 2749 | 7 | 174.279 | 49.484 | 0.284 | 2.669 | 1.394 | 0.006 |
| 30,666 | 3219 | 8 | 260.148 | 74.230 | 0.285 | 2.704 | 1.429 | 0.007 |
| 35,995 | 3779 | 9 | 370.405 | 106.114 | 0.286 | 2.765 | 1.491 | 0.007 |
| 39,616 | 4159 | 10 | 508.101 | 145.507 | 0.286 | 2.751 | 1.477 | 0.008 |

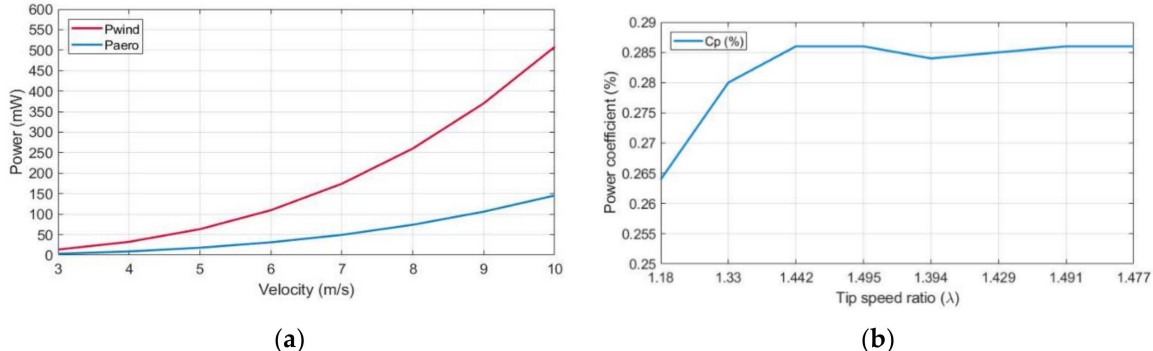

(**a**)      (**b**)

**Figure 6.** Open space system values. (**a**) The power difference between $P_{wind}$ and $P_{aero}$ at different wind velocities. (**b**) Aerodynamic efficiency at the same ranges.

In this case, the mechanical design seems to be unsuitable for this purpose. $P_{wind}$ and $P_{aero}$ are sometimes lower than achieved results on the test. Moreover, the electrical efficiency will be near zero as Table 3 shows. However, despite these results, it was decided to continue with the calculation [36] of the electric power harvested to verify whether or not the theory was valid for this system.

After the harvester captures the wind kinetic energy, the rotor rotates at a specific angular velocity under the drive torque, as in Equation (8).

$$T_{drive} = \frac{P_{aero}}{\omega} = \frac{\rho v_1^3 A C_p(\lambda,\theta)}{2\omega} \tag{8}$$

The drive torque ($T_{drive}$) is proportional to the cube of the wind velocity, power coefficient and inversely proportional to the angular velocity of the rotor, $\omega$. Moreover, when the harvester rotates at a low tip speed ratio, the ratio of $C_p = \omega$ remained mostly constant. Therefore, the drive torque is expressed in Equation (8) and could be treated as a constant if the ambient wind speed remains constant. For this system, the values obtained from $T_{drive}$ are shown in Table 4 and Figure 7.

**Table 4.** Numerical values of the torque vs. angular velocity.

| Angular Velocity (rad/s) | $T_{drive}$ (mW/rad/s) |
|---|---|
| 997 | 0.0036 |
| 1498.51 | 0.0061 |
| 2031.14 | 0.0089 |
| 2526 | 0.0124 |
| 2749 | 0.0180 |
| 3219.99 | 0.0231 |
| 3779.50 | 0.0281 |
| 4159.75 | 0.0350 |

**Figure 7.** Torque vs. angular velocity.

The electro-mechanical coefficient, $G$ (Volt/rad/s), is obtained with a linear relationship between the generated voltage, $V$ (V), and the angular velocity, $\omega$ (rad/s): Equation (9).

$$G = \frac{V}{\omega} \tag{9}$$

Electro-mechanical coefficient results for different air velocities (3–10 m/s) are presented in Table 5 and Figure 8.

**Table 5.** Numerical values of the electro-mechanical coefficient for different air velocities.

| Velocity (m/s) | Voltage (V) | $\omega$ (rad/s) | $G$ (Volt/rad/s) |
|---|---|---|---|
| 3 | 2.05 | 997 | 0.00205 |
| 4 | 3.02 | 1498.51 | 0.00202 |
| 5 | 4.17 | 2031.14 | 0.00205 |
| 6 | 5.32 | 2526 | 0.00211 |
| 7 | 5.98 | 2749 | 0.00218 |
| 8 | 6.66 | 3219.99 | 0.00207 |
| 9 | 7.57 | 3779.50 | 0.00200 |
| 10 | 8.38 | 4159.75 | 0.00202 |

Then, harvested power is calculated at load with Equation (10):

$$P_{e\_load} = \frac{G^2 \omega^2 R_L}{(R_L + R_{in})^2} \tag{10}$$

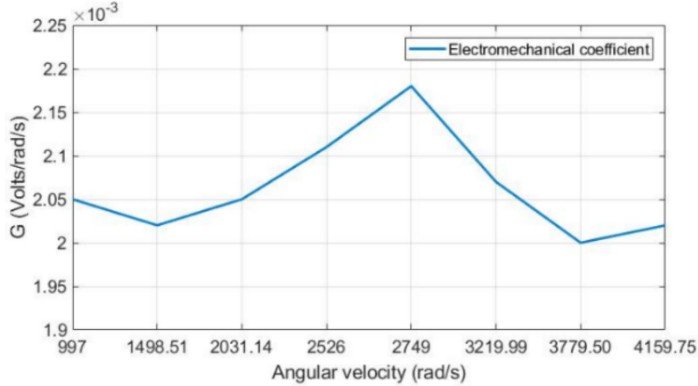

**Figure 8.** Electro-mechanical coefficient for different air velocities.

The result of harvested power for different loads (1–36 Ω) and air velocities (3–10 m/s) are shown in Table 6 and Figure 9:

**Table 6.** Calculated harvester power values for different loads and air velocities.

| | $P_{e\_load}$ (mW) | | | | | | | | | | | | |
|---|---|---|---|---|---|---|---|---|---|---|---|---|---|
| Velocity (m/s) | 1 Ω | 4 Ω | 7 Ω | 10 Ω | 13 Ω | 16 Ω | 19 Ω | 22 Ω | 25 Ω | 28 Ω | 31 Ω | 34 Ω | 36 Ω |
| 3 | 0.029 | 0.074 | 0.090 | 0.095 | 0.095 | 0.092 | 0.088 | 0.085 | 0.081 | 0.077 | 0.074 | 0.070 | 0.068 |
| 4 | 0.063 | 0.163 | 0.198 | 0.207 | 0.206 | 0.201 | 0.193 | 0.185 | 0.176 | 0.168 | 0.161 | 0.153 | 0.149 |
| 5 | 0.121 | 0.309 | 0.376 | 0.394 | 0.392 | 0.382 | 0.367 | 0.351 | 0.335 | 0.320 | 0.305 | 0.292 | 0.283 |
| 6 | 0.196 | 0.503 | 0.611 | 0.641 | 0.638 | 0.621 | 0.597 | 0.571 | 0.545 | 0.520 | 0.497 | 0.475 | 0.461 |
| 7 | 0.248 | 0.636 | 0.773 | 0.811 | 0.807 | 0.785 | 0.755 | 0.723 | 0.690 | 0.658 | 0.629 | 0.600 | 0.583 |
| 8 | 0.308 | 0.789 | 0.959 | 1.006 | 1.002 | 0.974 | 0.937 | 0.897 | 0.856 | 0.817 | 0.780 | 0.745 | 0.723 |
| 9 | 0.398 | 1.018 | 1.237 | 1.299 | 1.293 | 1.257 | 1.209 | 1.157 | 1.105 | 1.054 | 1.007 | 0.962 | 0.933 |
| 10 | 0.488 | 1.250 | 1.519 | 1.594 | 1.587 | 1.543 | 1.484 | 1.420 | 1.356 | 1.294 | 1.235 | 1.180 | 1.146 |

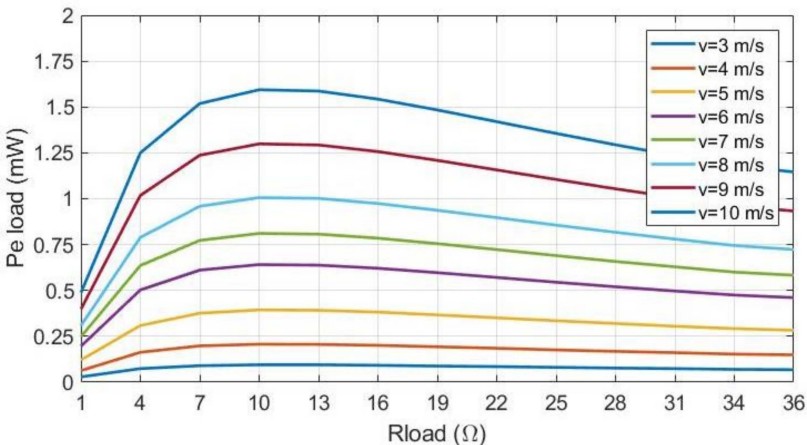

**Figure 9.** Calculated harvester power values for different loads and air velocities.

Finally, electric losses dependent on internal resistance ($R_{in}$) are given by the Equation (11):

$$P_{e\_load} = \frac{G^2 \omega^2 R_{in}}{(R_L + R_{in})^2} \tag{11}$$

The results of electrical power losses for 1–36 Ω and 3–10 m/s ranges are shown in Table 7 and Figure 10.

**Table 7.** Calculated harvester electric losses values for different loads and air velocities.

| | $P_{e\_loss}$ (mW) | | | | | | | | | | | |
|---|---|---|---|---|---|---|---|---|---|---|---|---|
| Velocity (m/s) | 1 Ω | 4 Ω | 7 Ω | 10 Ω | 13 Ω | 16 Ω | 19 Ω | 22 Ω | 25 Ω | 28 Ω | 31 Ω | 34 Ω | 36 Ω |
| 3 | 0.320 | 0.205 | 0.142 | 0.104 | 0.080 | 0.063 | 0.051 | 0.042 | 0.036 | 0.030 | 0.026 | 0.023 | 0.021 |
| 4 | 0.698 | 0.447 | 0.310 | 0.228 | 0.175 | 0.138 | 0.112 | 0.092 | 0.092 | 0.06 | 0.057 | 0.050 | 0.046 |
| 5 | 1.328 | 0.850 | 0.590 | 0.434 | 0.332 | 0.262 | 0.212 | 0.176 | 0.176 | 0.126 | 0.108 | 0.094 | 0.087 |
| 6 | 2.160 | 1.382 | 0.960 | 0.705 | 0.540 | 0.427 | 0.346 | 0.286 | 0.240 | 0.204 | 0.176 | 0.154 | 0.141 |
| 7 | 2.732 | 1.748 | 1.214 | 0.892 | 0.683 | 0.540 | 0.437 | 0.361 | 0.304 | 0.259 | 0.223 | 0.194 | 0.178 |
| 8 | 3.390 | 2.170 | 1.507 | 1.107 | 0.848 | 0.670 | 0.542 | 0.448 | 0.377 | 0.321 | 0.277 | 0.241 | 0.221 |
| 9 | 4.375 | 2.800 | 1.945 | 1.429 | 1.094 | 0.864 | 0.700 | 0.579 | 0.486 | 0.414 | 0.357 | 0.311 | 0.285 |
| 10 | 5.370 | 3.437 | 2.387 | 1.754 | 1.343 | 1.1 | 0.859 | 0.710 | 0.597 | 0.508 | 0.438 | 0.382 | 0.350 |

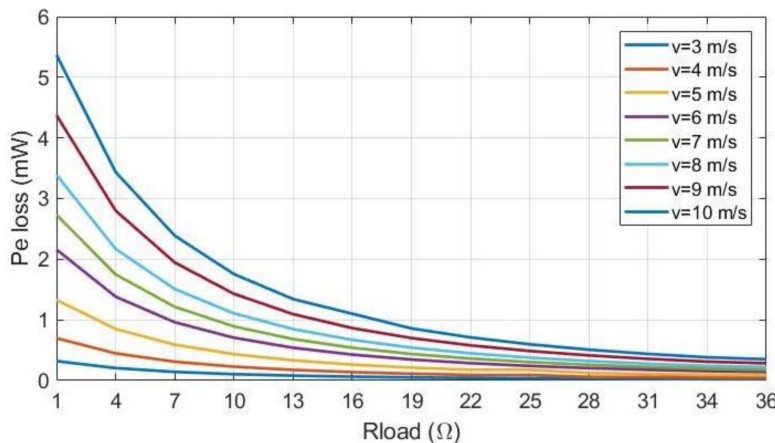

**Figure 10.** Calculated harvester electric losses for different loads and air velocities.

The analysis of the mathematical results shows that there are some states that are unfeasible, such as $P_{wind} < P_{electric}$ and $P_{aero} < P_{electric}$, because they do not comply with the law of conservation of energy. To further corroborate the mathematical results and discard error in the calculations, four tests were performed in open space conditions.

A quick experiment was carried out in ambient open space to verify the described basics. The experiment was designed with two variables: the wind velocity and the resistive load. These variables were considered due to the results achieved with the theoretical calculus and the minimum range of wind velocity used in other works [38].

A fan with controllable wind velocity levels, an anemometer to measure the wind velocity, the mini-turbine to analyze and an oscilloscope to measure generated signals and energy levels were used for open space tests, as can be seen in Figure 11.

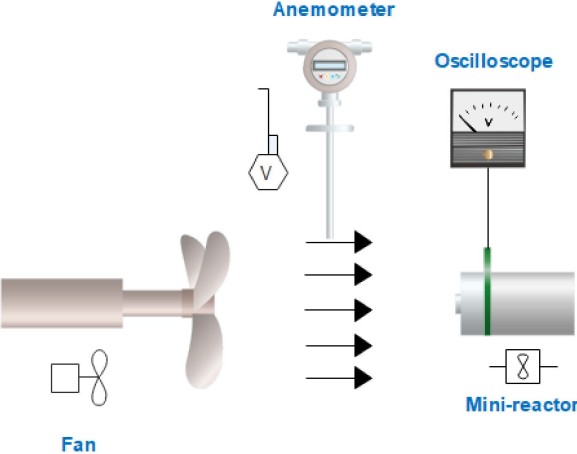

**Figure 11.** Test set-up diagram to measure wind power in open space for turbine model.

Four tests were carried out:

- Wind velocity → 3 m/s and 4 m/s
- Resistive load → 36 Ω and 1e6 Ω ($1 \times 10^6$ Ω)

Table 8 and Figure 12 show the values obtained from the tests. These results are similar to those calculated with the mathematical equation of the described operation principle.

**Table 8.** Open space test to verify harvester operation mode.

|  | $v = 3$ m/s | | $v = 4$ m/s | |
|---|---|---|---|---|
| $R_{load}$ (Ω) | 36 | $1 \times 10^6$ | 36 | $1 \times 10^6$ |
| Voltage ($V_{pp}$) | 0.06 | 0.152 | 0.116 | 0.28 |
| $P_{electric}$ (mW) | 0.037 | $8.66 \times 10^{-6}$ | 0.14 | $2.95 \times 10^{-5}$ |
| $P_{wind}$ (mW) | 13.718 | | 32.518 | |
| $P_{aero}$ (mW) | 0.389 | 0.473 | 1.019 | 1.295 |
| $\lambda$ | 0.043 | 0.088 | 0.066 | 0.124 |
| $C_p(\lambda,\theta)$ | 0.028 | 0.034 | 0.031 | 0.039 |
| Efficiency (%) | 0.269 | $6.31 \times 10^{-6}$ | 0.43 | $9.07 \times 10^{-5}$ |

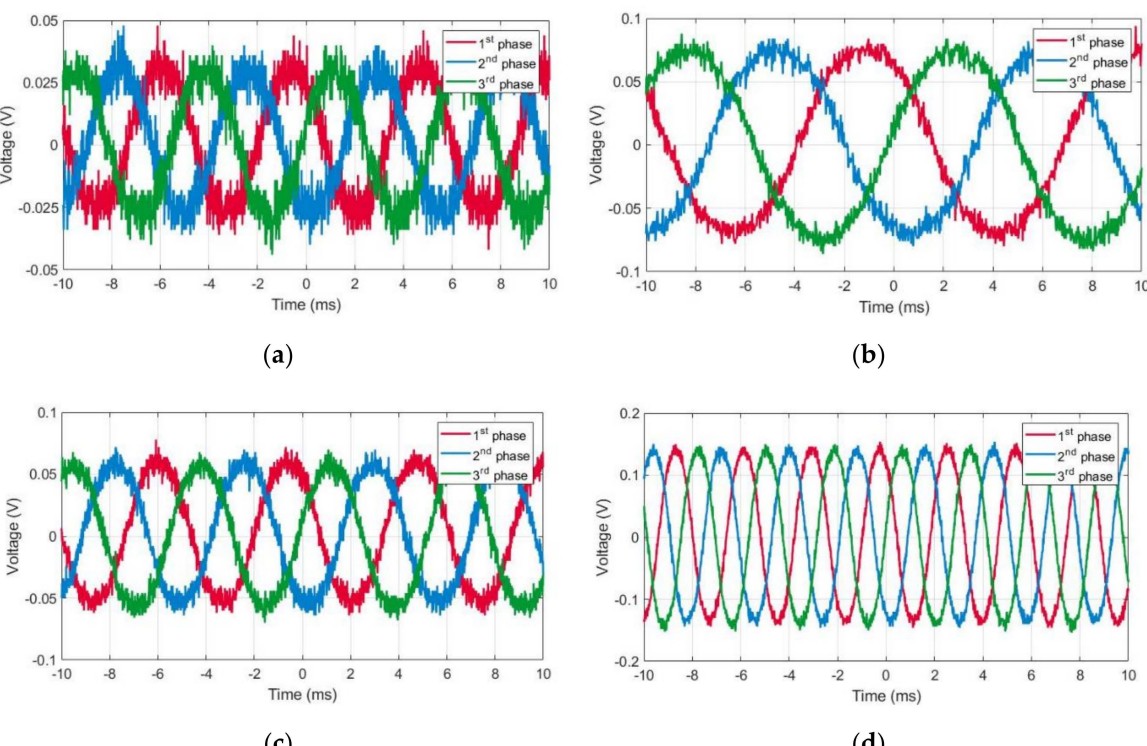

**Figure 12.** Measured signals of the harvester. (**a**) A 3 m/s air velocity in open space for a 36 Ω load. (**b**) A 3 m/s air velocity in open space for a $10^6$ Ω load. (**c**) A 4 m/s air velocity in open space for a 36 Ω load. (**d**) A 4 m/s air velocity in open space for a $10^6$ Ω load.

The achieved results are in concordance with the results of the calculation, those being the theoretically calculated and the measured output power, for 3 m/s wind velocity, 0.068 mW and 0.037 mW, respectively, and for 4 m/s wind velocity, 0.149 mW and 0.14 mW, respectively.

The conclusion for both, the calculated and measured results, is that the Equation (4) is not adequate to calculate wind power in a pipe. The main reason is that the environment is different, and the pressure parameter must be considered to achieve the correct value of wind power through the harvester.

### 3.2.2. Work Operation inside a Closed Environment

Figure 13 shows the test set-up diagram, which has the following elements: an air pump, a gate valve, a control valve with a gauge, a connection pipe, two pitot tubes, an anemometer, an oscilloscope, one test-bench, and the mini wind harvester.

The test set-up works as follows: the air pump provides the required air flow, whose pressure gate valve controls its insertion. The pitot tubes measure the pressure value before and after the mini wind harvester, and the anemometer provides the air velocity. In addition, the oscilloscope is used to collect the data for post-processing.

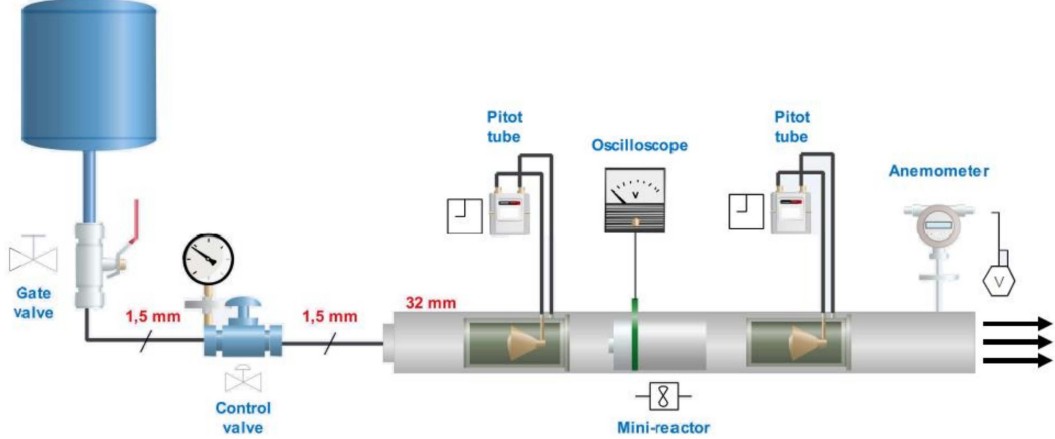

**Figure 13.** Test set-up diagram to measure wind power for the mini wind harvester model.

Table 9 shows the results of tests carried out to measure the pressure level for the different wind velocities. The table also provides the equivalent wind power for its corresponding pressure level.

**Table 9.** Measured wind power values for different wind flow ranges.

| $\gamma$ (g/cm$^3$) | A (cm$^2$) | v (cm/s) | Q (cm$^3$/s) | H (mbar) | Wind Power (mW) |
|---|---|---|---|---|---|
| 0.0013 | 8.04 | 300 | 2412 | 20 | 62.712 |
| 0.0013 | 8.04 | 400 | 3216 | 36 | 150.509 |
| 0.0013 | 8.04 | 500 | 4020 | 52 | 271.752 |
| 0.0013 | 8.04 | 600 | 4824 | 91 | 570.679 |
| 0.0013 | 8.04 | 700 | 5628 | 112 | 819.437 |
| 0.0013 | 8.04 | 800 | 6432 | 142 | 1187.347 |
| 0.0013 | 8.04 | 900 | 7236 | 144 | 1354.579 |
| 0.0013 | 8.04 | 1000 | 8040 | 163 | 1703.676 |

Where $\gamma$ is the density of the air, A is the effective area of the wind harvester, v is the wind velocity, Q is the volumetric flow rate, and H is the pressure inside the pipe.

With the wind power data for each flow velocity, the efficiencies, powers and losses of the harvester are obtained for the same test loads used in this work, in order to characterize the behavior of the different parts of the system.

Table 10 and Figure 14 show the results for loads of 27 Ω, 11 Ω and 100 Ω. These values are corresponded to the maximum efficiency and produced electrical power, thus, the lower mechanical losses. The best results are achieved for an impedance matching load of 27 Ω. In the 11 Ω load condition, the load impedance is mismatched with the source, the mechanical losses increase, and the efficiency of the system begins to decrease. Finally, with a 100 Ω load, unless the energy demand is lower, there is higher impedance mismatching, and the mechanical losses increase more than in the previous load conditions. The efficiency of the system does not reach the 11%.

**Table 10.** Generated electric power, mechanical losses, and system efficiency values from 3 m/s to 10 m/s; from 20 mbar to 163 mbar with 27, 11 and 100 Ω loads.

| v (m/s) | Pressure (mbar) | Wind Power (mW) | Rload (Ω) | Elec. Power (mW) | Mech. Losses (mW) | Efficiency (%) |
|---|---|---|---|---|---|---|
| 3 | 20 | 62.712 | | 15.284 | 47.428 | 24.37 |
| 4 | 36 | 150.509 | | 31.155 | 119.354 | 20.70 |
| 5 | 52 | 271.752 | | 55.489 | 216.263 | 20.42 |
| 6 | 91 | 570.679 | 27 | 111.023 | 459.657 | 19.45 |
| 7 | 112 | 819.437 | | 160.716 | 658.721 | 19.61 |
| 8 | 142 | 1187.347 | | 239.686 | 947.661 | 20.19 |
| 9 | 144 | 1354.579 | | 272.543 | 1082.036 | 20.12 |
| 10 | 163 | 1703.676 | | 333.146 | 1370.530 | 19.55 |
| 3 | 20 | 62.712 | | 15.284 | 47.428 | 24.37 |
| 4 | 36 | 150.509 | | 31.155 | 119.354 | 20.70 |
| 5 | 52 | 271.752 | | 55.489 | 216.263 | 20.42 |
| 6 | 91 | 570.679 | 11 | 111.023 | 459.657 | 19.45 |
| 7 | 112 | 819.437 | | 160.716 | 658.721 | 19.61 |
| 8 | 142 | 1187.347 | | 239.686 | 947.661 | 20.19 |
| 9 | 144 | 1354.579 | | 272.543 | 1082.036 | 20.12 |
| 10 | 163 | 1703.676 | | 333.146 | 1370.530 | 19.55 |
| 3 | 20 | 62.712 | | 15.284 | 47.428 | 24.37 |
| 4 | 36 | 150.509 | | 31.155 | 119.354 | 20.70 |
| 5 | 52 | 271.752 | | 55.489 | 216.263 | 20.42 |
| 6 | 91 | 570.679 | 100 | 111.023 | 459.657 | 19.45 |
| 7 | 112 | 819.437 | | 160.716 | 658.721 | 19.61 |
| 8 | 142 | 1187.347 | | 239.686 | 947.661 | 20.19 |
| 9 | 144 | 1354.579 | | 272.543 | 1082.036 | 20.12 |
| 10 | 163 | 1703.676 | | 333.146 | 1370.530 | 19.55 |

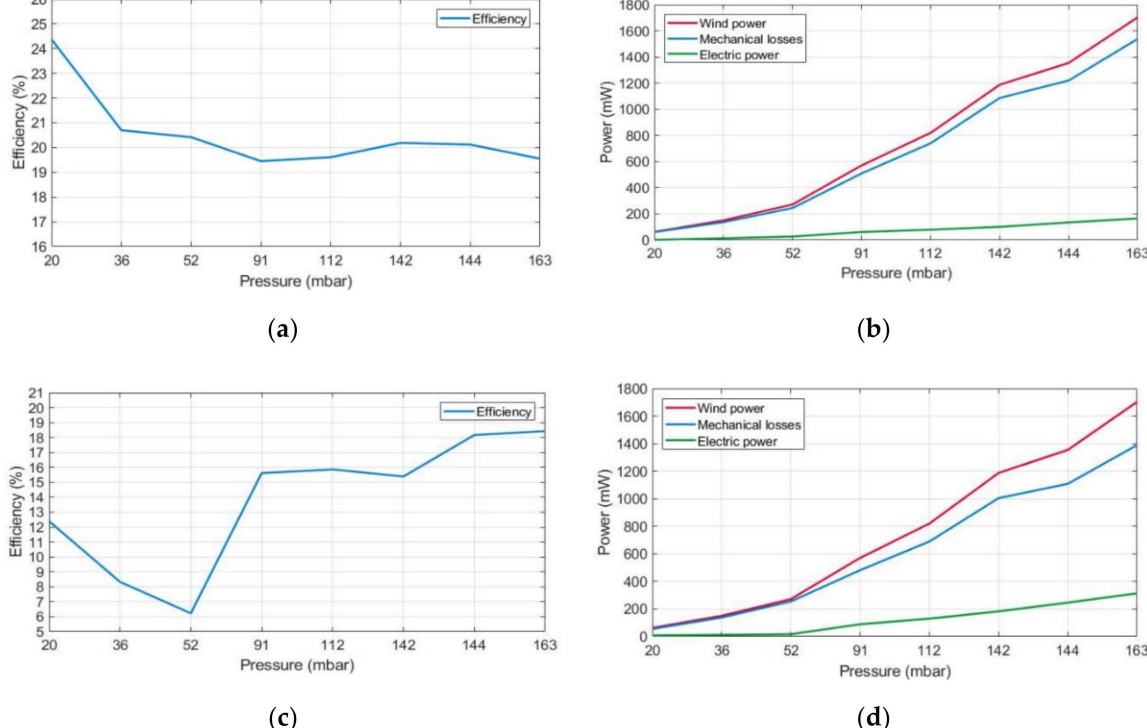

**Figure 14.** *Cont.*

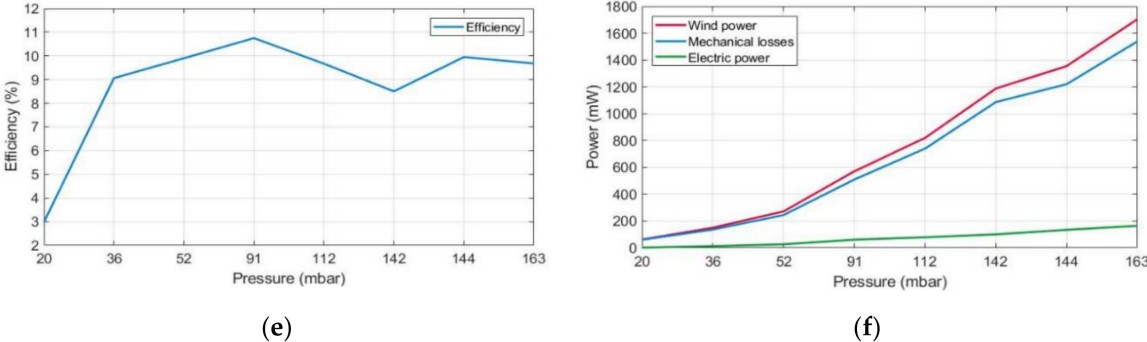

(**e**)  (**f**)

**Figure 14.** Generated values of electric power, mechanical losses and system efficiency values from 3 m/s to 10 m/s; from10 mbar to 163 mbar with 27 Ω (**a,b**), 11 Ω (**c,d**) and 100 Ω load (**e,f**). (**a**) System efficiency (27 Ω). (**b**) Power differences/losses through different steps of the mini wind harvester (27 Ω). (**c**) System efficiency (11 Ω). (**d**) Power differences/losses through different steps of the mini wind harvester (11 Ω). (**e**) System efficiency (100 Ω). (**f**) Power differences/losses through different steps of the mini wind harvester (100 Ω).

The presented measures of voltage, power and frequency were obtained for the impedance matching load and a varying wind flow velocity between 3 m/s and 10 m/s. However, it is likely that the mini wind harvester with 2 m/s wind velocity should be able to produce enough power.

Although the harvested energy is a direct function of the pressure energy, several graphs and data are expressed as a function of the velocity of the air flow and not as a function of the pressure because in m/s the scale is linear. It is easier to interpret a linear scale of velocity between 3–10 m/s than a non-linear scale of pressure between 62–1703 mW. However, Table 9 could be used to make the conversion.

Moreover, the effect of the impedance matching condition to obtain the maximum power in each condition was analyzed, using resistors as the electric load (previous experiments have been carried out considering this effect). In order to search for the impedance matching condition, the torque or momentum force effect must be analyzed; this is produced when an electric load is connected to the generator. This electrical load changes system global resistance; consequently, the magnetic field produced by magnets is also altered, due to their linear relationship. It gives as a result a reduction of the angular velocity due to the brake effect that happens in the system.

The system torque results achieved with nine different resistors are presented in Figure 15a and in Table 11. Then, Figure 15b and Table 12 show the power generation and system linearity as a function of fluid velocity and resistive loads. The validation of the principle of maximum power achievement when the torque value is at its maximum is also accomplished.

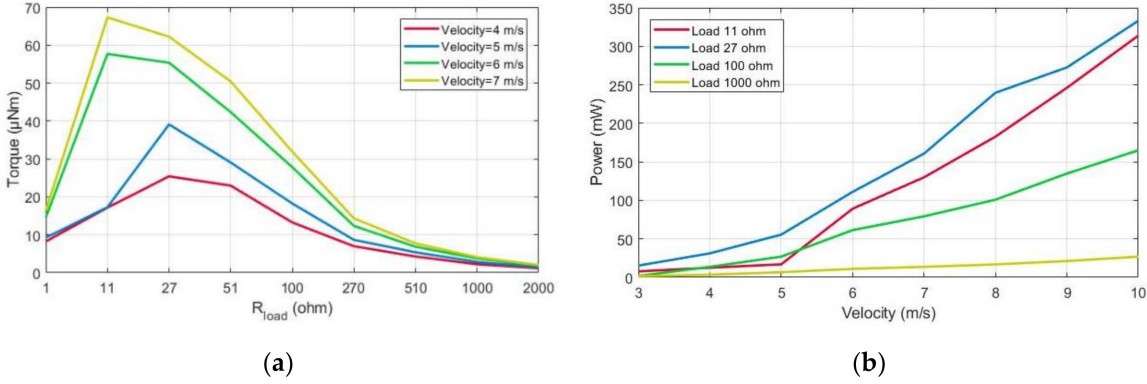

(**a**)  (**b**)

**Figure 15.** Measured operational values at different flow velocities and loads. (**a**) Torque representation at different flow velocities. (**b**) Measured power values.

**Table 11.** Measured values for torque representation.

| Load (Ω) | Torque (μNm) | | | |
|---|---|---|---|---|
| | Velocity (m/s) | | | |
| | 4 | 5 | 6 | 7 |
| 1 | 8.26 | 9.27 | 14.55 | 16.37 |
| 11 | 17.15 | 17.25 | 57.67 | 67.30 |
| 27 | 25.41 | 39.10 | 55.37 | 62.23 |
| 51 | 22.97 | 29.06 | 42.34 | 50.43 |
| 100 | 13.26 | 18.25 | 27.83 | 31.93 |
| 270 | 6.99 | 8.67 | 12.35 | 14.33 |
| 510 | 4.28 | 5.39 | 6.85 | 7.74 |
| 1000 | 2.23 | 2.83 | 3.74 | 4.10 |
| 2000 | 1.20 | 1.46 | 1.86 | 2.02 |

**Table 12.** Measured power at different conditions with the three-phase generator.

| Velocity (m/s) | Power (mW) | | | |
|---|---|---|---|---|
| | Resistor Load (Ω) | | | |
| | 1 | 27 | 11 | 100 |
| 3 | 1.61 | 15.28 | 7.77 | 1.87 |
| 4 | 3.45 | 31.16 | 12.54 | 13.64 |
| 5 | 6.72 | 55.49 | 16.93 | 26.91 |
| 6 | 11.04 | 111.02 | 89.15 | 61.34 |
| 7 | 13.68 | 160.72 | 129.94 | 79.25 |
| 8 | 16.80 | 239.69 | 182.70 | 100.87 |
| 9 | 21.21 | 272.54 | 246.13 | 134.79 |
| 10 | 26.86 | 333.15 | 313.92 | 164.98 |

Finally, Table 13 presents a summary and quantitative comparison of the discussed wind harvested technologies in the introduction section with the developed harvester in this work.

**Table 13.** Comparison of the performances of wind energy harvesters.

| Ref | Size (mm) | Voltage (V) | Wind Velocity (m/s) | Max Output Power (mW) | Cp (%) | Power Density Per Area (mW × cm²) | Topology |
|---|---|---|---|---|---|---|---|
| [20] | 10 ø | 80 | 2.7 | 18.3 | - | 0.58 | Electrostatic |
| [19] | 40 ø | 450 | 10 | 1.8 | 0.24 | 0.143 | Electrostatic |
| [17] | 161 × 250 | 30 | 5.8 | 53 | - | 0.13 | Piezoelectric |
| [22] | 30 × 12.5 | 14 | 160.2 | 2.56 | - | 0.68 | Piezoelectric |
| [24] | 70 × 20 | 7.86 | 14 | 0.061 | - | 0.004 | Piezoelectric |
| [17] | 490 × 20 | 6 | 7 | 70 | 3.46 | 0.71 | Electromagnetic |
| [39] | 26 ø | 2.2 | 10 | 7 mW | 11 | 1.07 | Electromagnetic |
| [40] | 42 ø | 2.6 | 11.8 | 130 | - | 2.34 | Electromagnetic |
| This work | 32 ø | 2.45 | 10 | 333 | 28 | 10.32 | Electromagnetic |

*3.3. Electro-Mechanical Equivalent Model of the Mini Wind Harvester or Reactor*

3.3.1. Equivalent Model Description

Figure 16 provides the mixed behavioral and structural–electric model of the mini wind harvester. The wind energy and the conversion of energy to mechanical and electrical is modeled through mathematical equations, voltage and controlled current sources and passive components. The basis of this equivalent circuit were taken from reference [36], and the values were adapted to the wind harvester presented in this work and the circuit to a three-phase system.

$I_{torque}$ is the equivalent electrical representation of the mechanical torque of the wind harvester, $V_{electric1,2,3}$ and $I_{gen1,2,3}$ are the electric voltage and current generated by each phase coil, and $V_{out1,2,3}$ are the voltage produced by the coils but with the corresponding phase shift, 0°, 120° and 240°.

The equivalent model presented is based on different equations and elements equivalent to the harvester work operation principles and the electro-mechanical characteristics of the system [37]. The following lines describe the three parts that constitute the circuit.

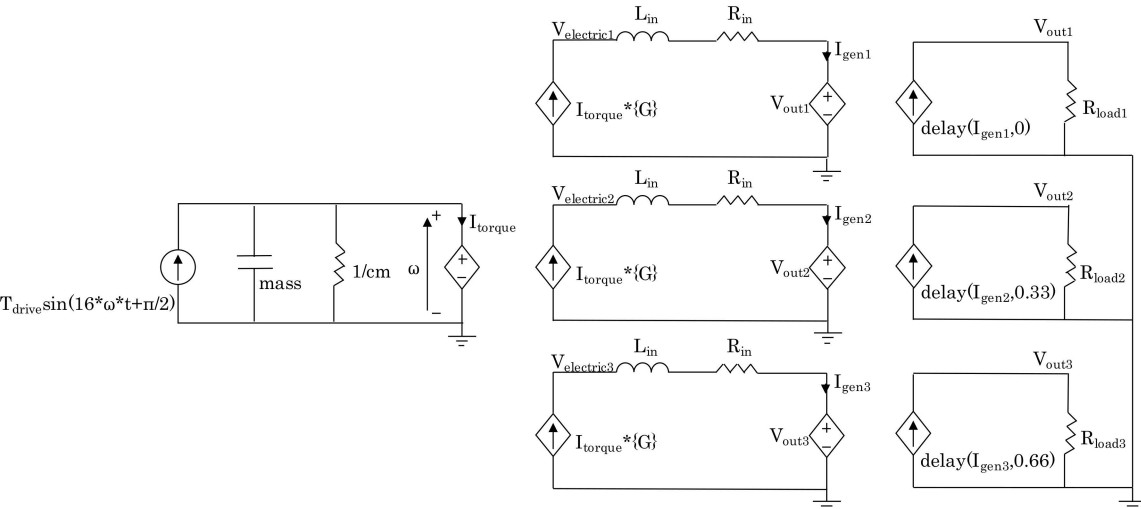

**Figure 16.** Schematic of mini wind harvester model.

The first part of the schematic includes the wind and mechanical elements. Wind power is characterized by the amplitude of the input sinusoidal current source. The drive torque ($T_{drive}^*$) is deduced as Equation (12) of [41].

$$T_{drive}^* = m\dot{\omega} + \omega \left[ \frac{c_m + G^2}{R_{in} + R_L} \right] \tag{12}$$

where $\dot{\omega}$ is the angular acceleration, $\omega$ is the angular velocity, $c_m$ is the coefficient of frictional torque and it is represented with a resistance, $R_{in}$ is the inherent resistance of the generator, $G$ is the electro-mechanical coefficient and m is the mass of the rotational parts of the mini wind harvester. The mass $m$ is represented with a capacitor because the differential equation that characterizes the movement of a weight or inertial mass is equivalent to the differential equation of an Resistor-Inductor-Capacitor (RLC) network. Nevertheless, when the flow has a constant pressure/velocity, the mini wind harvester operates at a constant angular velocity condition. Consequently, the angular acceleration becomes zero. Therefore, Equation (12) is rewritten as (13):

$$T_{drive}^* = \omega \left[ \frac{c_m + G^2}{R_{in} + R_L} \right] \tag{13}$$

The equivalence between wind power and harvested mechanical power is represented with (14):

$$P = V \cdot I = \omega \cdot \tau \tag{14}$$

where $P$ is the power, $V$ is the voltage, $I$ is the current, $\omega$ is the angular velocity, and $\tau$ is the harvester torque or the momentum force (the product of the turbine mass and the velocity of the wind). This representation of angular velocity and torque of the mini wind harvester is translated to the circuit as a voltage source and the current which pass through the source.

Angular velocity and torque are functions of the wind power and the resistive load of the system, which leads to a complex relationship between both magnitudes. However, it could be stated that the drive torque is directly proportional to the angular velocity under a constant resistive load condition.

Once the mechanical equivalence of the circuit has been described, the conversion of mechanical energy to electromagnetic (thanks to the rotatory movement of the magnets) and the magnetic field losses or the binding between mechanical and electric parts is represented with the electro-mechanical coefficient, $G$.

The next block of the model represents the electric side. It has three equal circuits, one for each secondary. $R_{in}$ and $L_{in}$ are the internal resistance of the PCB and the induction value of the designed and manufactured coils (the represented $R_{in}$ and $L_{in}$ in Figure 16 are the sum of the 16 series coils' resistance and the coil value of each phase).

In the case of the present study, the inductance effect in the microgenerator $L_{in}$ can be neglected in the calculation of the output voltage. Consequently, the output voltage of each phase is written as (15).

$$V_{out} = \frac{G\omega R_L}{(R_{in} + R_L)} \tag{15}$$

The last step provides the correct phase-shift of each phase. It is implemented with delayed voltage controlled current sources with a conversion ratio of one to one.

### 3.3.2. Parameters Measurement and Obtainment

The DC $R_{in}$ was measured with a polymeter and is 11 $\Omega$ for all the phases.

The electro-mechanical coefficient, $G$ (V/rad/s), is obtained as a linear relationship between the generated voltage, $V$ (V), and the angular velocity, $\omega$ (rad/s) as shown in Equation (16):

$$G = \frac{V}{\omega} \tag{16}$$

Electro-mechanical coefficients for different air velocities ranging from 3 to 10 m/s are presented in Table 14. The most suitable coefficient is 0.002 V/rad/s:

**Table 14.** Electro-mechanical coefficient for different air velocities.

| Velocity (m/s) | Voltage (V) | $\omega$ (rad/s) | $G$ (V/rad/s) |
|---|---|---|---|
| 3 | 2.05 | 997 | 0.00205 |
| 4 | 3.02 | 1498.51 | 0.00202 |
| 5 | 4.17 | 2031.14 | 0.00205 |
| 6 | 5.32 | 2526 | 0.00211 |
| 7 | 5.98 | 2749 | 0.00218 |
| 8 | 6.66 | 3219.99 | 0.00207 |
| 9 | 7.57 | 3779.50 | 0.00200 |
| 10 | 8.38 | 4159.75 | 0.00202 |

The damping effect constitutes a mechanical element in the electro-mathematical model. This behavior is represented in the model as a resistance with a value given by Equation (17).

$$R_{damping} = \frac{1}{c_m} \tag{17}$$

where $c_m$ is the damping coefficient. The aerodynamic damping can be determined using a non-linear time domain equation. However, the damping $\delta$ (aerodynamic plus structural damping) can also be calculated from the logarithmic decrement, as shown in Equation (18):

$$R_{damping} = \delta = \frac{1}{n} \log\left(\frac{x_{n-p}}{x_n}\right) \tag{18}$$

This relationship only has meaning for underdamped systems because the logarithmic decrement is defined as the natural log of the ratio of any two successive amplitudes, and only underdamped

systems exhibit oscillation. As this system belongs to the latter group, consequently, Equation (18) could be used for damping coefficient calculation.

One technique to derive the aerodynamic damping ratio is to analyze the decay of the free vibration of the tower top after pulse loading. First, the mini wind harvester is set to normal work operation. When this operation mode is reached, the mini wind harvester is braked, and the voltage waveform at the output of the generator provides the damping ratio of the wind harvester, Figure 17.

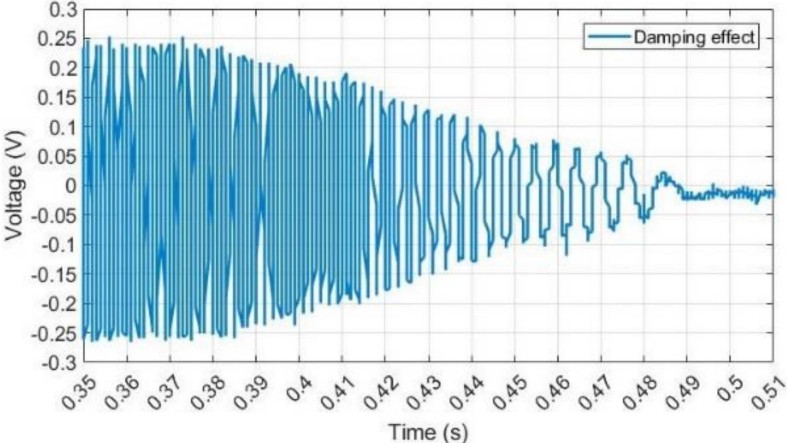

**Figure 17.** Damping ratio of mini wind harvester.

The accuracy of the data was improved by measuring several points which avoids any incongruent/fail result. Three different points were taken from the measured damper curve of Figure 17, besides, the reference points $x_0$, $x_2$, $x_6$, and $x_{10}$ would corroborate the similar value of the damping coefficient. Finally, Equation (18) was applied, achieving as a result values presented in Table 15.

**Table 15.** Coefficient of frictional torque calculation values.

| $x_0$ | $x_2$ | $\delta$ |
|-------|-------|----------|
| 0.169 | 0.153 | 0.0215 |

| $x_0$ | $x_6$ | $\delta$ |
|-------|-------|----------|
| 0.169 | 0.129 | 0.0195 |

| $x_0$ | $x_{10}$ | $\delta$ |
|-------|----------|----------|
| 0.169 | 0.0965 | 0.0243 |

The value chosen as the damping effect resistance is an intermedium value, thus: $R_{damping} = 1/c_m = 0.0215\ \Omega$.

### 3.4. Model and Test Results Comparison and Proposed System Verification

In order to verify the mini wind harvester proposed model, the simulations results were compared with the prototype measured values. Figure 18 summarizes the performance of the system for two load conditions, 10 Ω and 100 Ω, and two wind flow velocity conditions, 3 m/s, and 10 m/s. The load changes the output voltage and the frequency. If the load remains constant and the wind flow changes the system behaves the same. Thus, the decrease in the load or the wind flow leads to reduced output AC voltage amplitude and frequency.

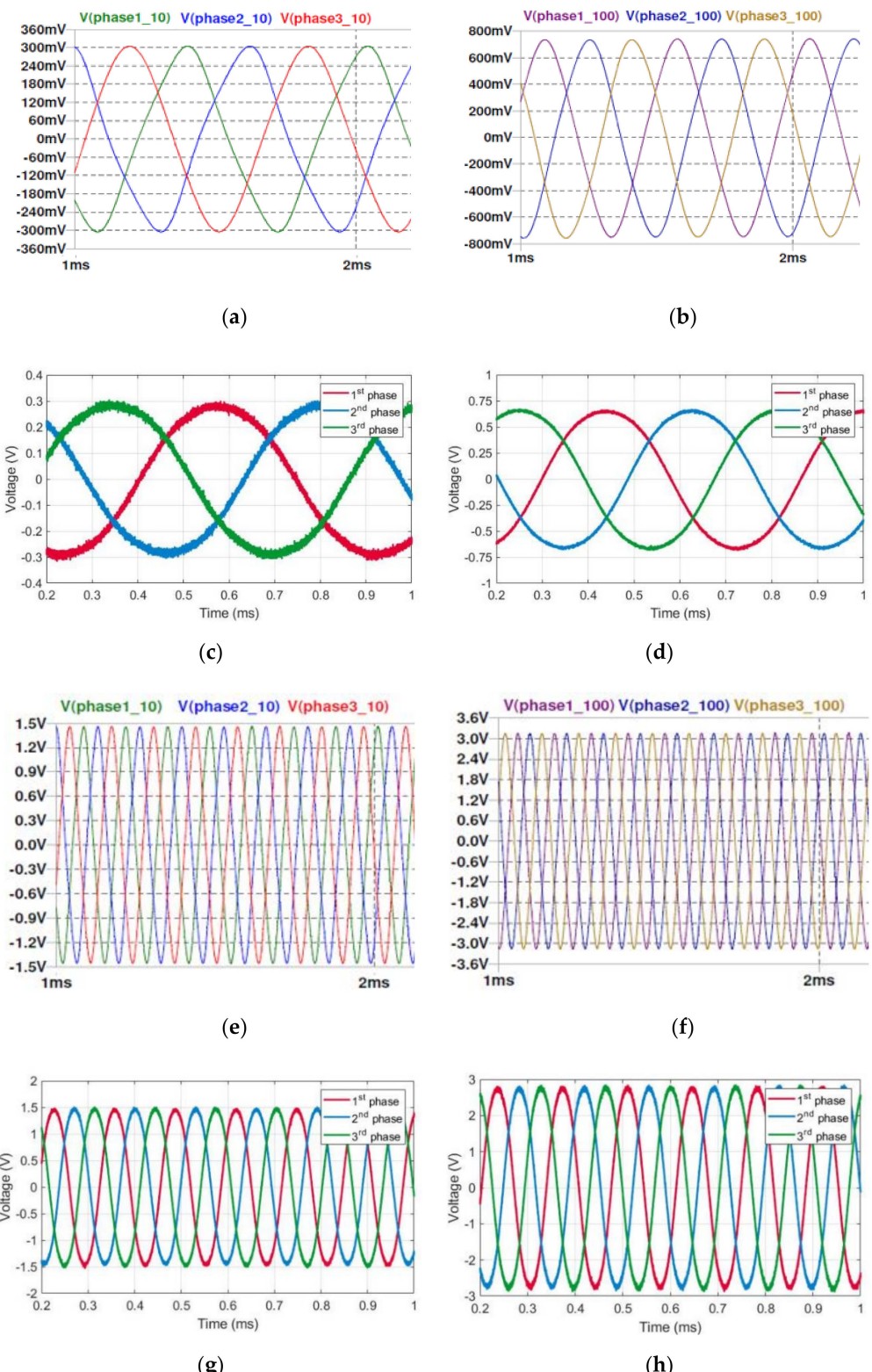

**Figure 18.** Linear Technology Spice (LTspice) model and real test results with 10 and 100 Ω loads and with 3 and 10 m/s wind flow velocities. (**a**) LTspice model results with 10 Ω loads and 3 m/s wind flow velocity. (**b**) LTspice model results with 100 Ω loads and 3 m/s wind flow velocity. (**c**) Real test results with 10 Ω load and 3 m/s wind flow velocity. (**d**) Real test results with 100 Ω load and 3 m/s wind flow velocity. (**e**) LTspice model results with 10 Ω loads and 10 m/s wind flow velocity. (**f**) LTspice model results with 100 Ω loads and 10 m/s wind flow velocity. (**g**) Real test results with 10 Ω load and 10 m/s wind flow velocity. (**h**) Real test results with 100 Ω load and 10 m/s wind flow velocity.

However, the mechanic limitations of the present prototype bearings set the maximum wind velocity to 10 m/s wind velocity. Table 16 and Figure 19 show a comparison of measures and simulations of the model under different wind flow powers with a load matching condition (27 $\Omega$).

**Table 16.** Comparison of results between the real measurements and model simulation under different wind conditions with a load impedance matching condition.

| | | | | | | | | | |
|---|---|---|---|---|---|---|---|---|---|
| | $R_{load} = 27\ \Omega$ | | | | | | | | |
| | **Results** | | | | | | **Statistics** | | |
| *v* (m/s) | **Real** | | | **Model** | | | **Relative Error (%)** | | |
| | $V_p$ (V) | $P_{out}$ (mW) | *f* (kHz) | $V_p$ (V) | $P_{out}$ (mW) | *f* (kHz) | $V_p$ | $P_{out}$ | *f* |
| 3 | 0.525 | 15.28 | 1.9 | 0.543 | 16.381 | 1.899 | 3.43 | 7.21 | 0.05 |
| 4 | 0.745 | 31.16 | 2.569 | 0.729 | 29.525 | 2.548 | 2.15 | 5.25 | 0.82 |
| 5 | 0.998 | 55.49 | 3.452 | 0.977 | 53.029 | 3.425 | 2.10 | 4.44 | 0.78 |
| 6 | 1.413 | 111.02 | 5.047 | 1.437 | 114.721 | 5.5 | 1.70 | 3.33 | 8.98 |
| 7 | 1.702 | 160.72 | 6.093 | 1.732 | 166.657 | 6.075 | 1.76 | 3.69 | 0.30 |
| 8 | 2.085 | 239.69 | 7.311 | 2.079 | 240.125 | 7.301 | 0.29 | 0.18 | 0.14 |
| 9 | 2.207 | 272.54 | 7.794 | 2.224 | 274.788 | 7.809 | 0.77 | 0.82 | 0.19 |
| 10 | 2.451 | 333.15 | 8.766 | 2.469 | 338.665 | 8.65 | 0.73 | 1.66 | 1.32 |

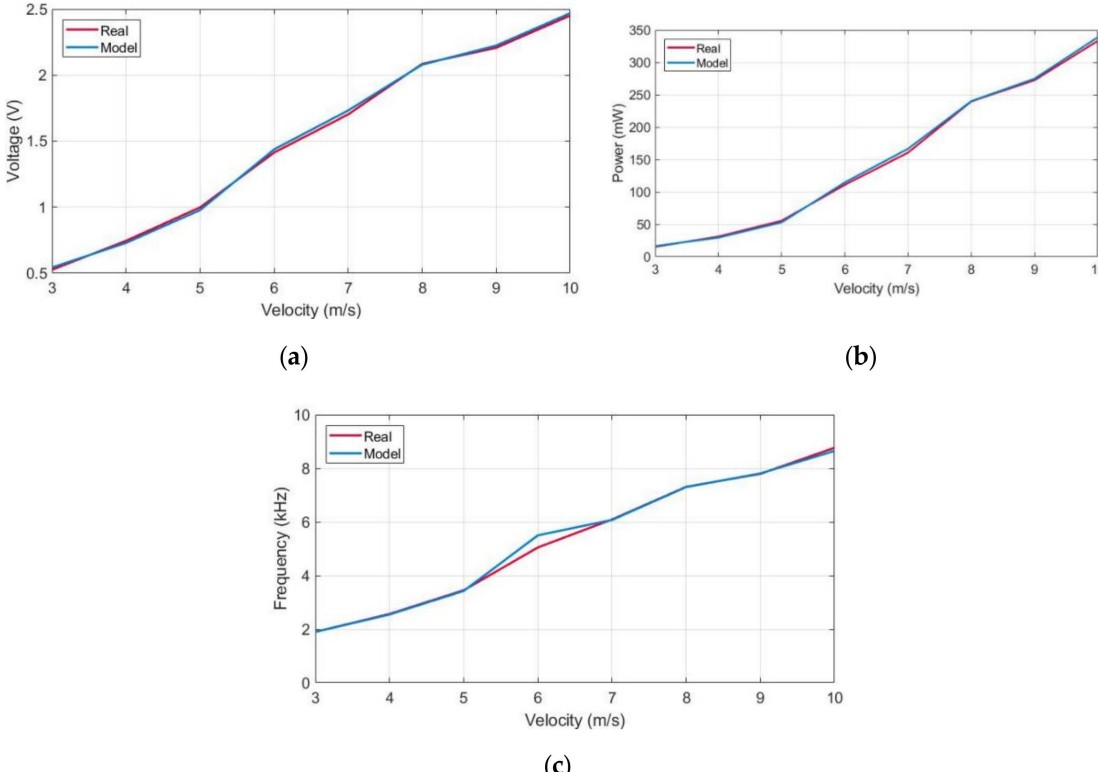

(a)

(b)

(c)

**Figure 19.** LTspice model vs. measured values comparison with a 27 $\Omega$ resistive load (impedance matching condition). (**a**) Simulated vs. measured voltage. (**b**) Simulated vs. measured power. (**c**) Simulated vs. measured power frequency.

The measurements have a maximum voltage and power relative error, for a wind velocity of 3 m/s, of 3.43% and 7.21%, respectively; regarding the frequency, the error is 8.98% for a wind velocity of 6 m/s, which is less overall. The obtained results confirm that the model met the requirements and can be used in the research, as long as the three results that have an error higher than 5% occurring only in

one of the values (voltage, power, and frequency) and not affecting the final results of the modelled equivalent circuit.

As a conclusion, if the harvester is installed inside a windy pipe, the absolute kinetic energy is zero, and the harvested power will be only the potential energy, as Equation (8) states. According to Bernoulli's principle and considering that the air is incompressible, wind velocities must be the same in both sides of the pipe, $v_1 = v_2$. In addition, due to the theorem of the energy principle, $P_1$ must be greater than $P_2$, $P_1 > P_2$, and consequently $\Delta P = P_1 - P_2$, and the harvested energy is given by Equation (19).

$$E_{harvested} = E_{potential1} - E_{potential2} \qquad (19)$$

where $E_{potential1}$ and $E_{potential2}$ are the potential energy of the wind before and after the harvester, and $E_{harvested}$ is the energy harvested by the harvester. The following hydraulic Equation (20) gives potential wind power:

$$P_{wind} = \gamma Q H \qquad (20)$$

where $\gamma$ is the specific weight of the air, $Q$ is the flow quantity across the pipe (A·v), and $H$ is the dynamic pressure difference between both sides. Electric power is achieved from wind potential power including mechanical power loses and it is calculated as in Equation (21):

$$P_{electric} = \Delta P_{wind} - P_{mechanic\_losses} \qquad (21)$$

where $\Delta P_{wind}$ is the power of the wind, $P_{mechanic\_losses}$ is the power losses of the harvester from the mechanical parts, and $P_{electric}$ is the generated electric power by the harvester.

So, the harvested energy corresponds to the potential energy of wind, not the kinetic energy. In addition, mechanical efficiency is achieved by subtracting air power with electrical power. Moreover, Betz's numbers are not applicable in this case as in the case of wind turbines. This law is only for turbines that harvest energy from wind velocity, i.e., from kinetic energy. For the describedreasons, i.e., this system harvests energy from available pressure and not from wind velocity; thus, the mini wind harvester is not a mini-turbine but a mini-reactor. Furthermore, the correct or appropriate environment to work for this harvester is inside a pipe.

## 4. AC/DC Converter

Harvested power is usually need in a DC regime. Consequently, a converter that changes an AC regime into DC must be included [42,43].

In this research the chosen conversion mode is the AC–DC direct conversion. Nevertheless, a basic architecture of direct rectification of an AC signal into DC will be analyzed too for comparison purposes: three phase diode bridge (Point 4.4.1).

Figure 20 shows the block diagram AC system with the harvester (mini wind harvester), power management (AC/DC converter) and the storage device (supercapacitor).

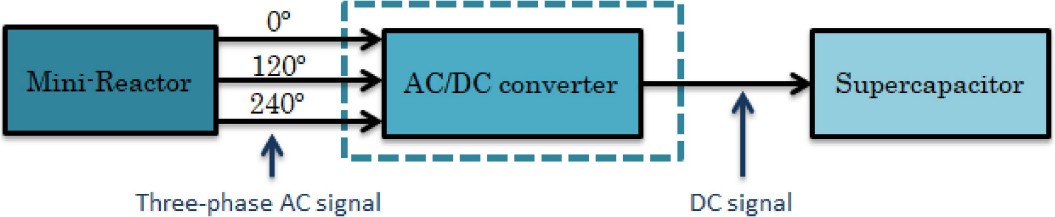

**Figure 20.** AC complete system block diagram.

The main objective of this subsection is to research, simulate and verify the feasibility of the development and implementation of an AC/DC converter for the mini wind harvester for voltage and frequency input ranges with more than a 50% efficiency.

However, the input operation principles are set by the mini wind harvester, which imposes several design requirements on the converter, such as:

- A low-value coil: only 2 µH. The harvester's own coil is used in order to avoid the losses of including another one.
- A low input AC voltage for energy harvesting the usual converters (voltage levels are between 0.3 V and 1.5 V, commercial diodes do not work properly in the rectification step with such low voltage levels due to the internal losses or the forward voltage, VF).
- An 8 kHz range frequency input: switching frequency changing.
- The influence of the reactive power of the generator must be considered.

On the other hand, electronic devices usually operate in the range of 3 V to 5 V. If the minimum voltage level is not reached, the voltage must be raised by a step-up device. On the contrary, if the voltage level exceeds the maximum, the power storage system will be damaged. Thus, decreasing the voltage level with a step-down device is required. In this research, these situations are not valid because an extra element is needed, not fulfilling the predefined requirements. Moreover, the converter must maintain its performance within the whole range of voltage and generator operating conditions. These aforementioned design requirements aim for a correct and efficient integration between the harvester and the converter.

The best AC/DC converter must be designed selecting the appropriate architecture for the usage described in this work; thus the architecture selection is shown after prior evaluation of diode and metal-oxide-semiconductor field-effect transistor (MOSFET) selection under different ambient conditions. All the analyses were carried out under two extreme test conditions, in order to guarantee that, in such conditions, the minimum and maximum points of energy and work conditions are achieved. As the system is mostly linear, the intermediate values are supposed to be linear and within the defined range:

1. A minimum wind velocity condition of 3 m/s, i.e., when available input power is the minimum possible, because at lower wind velocity the generator rotatory movement does not generate enough power to start working the electronic system.
2. A maximum wind velocity condition of 10 m/s, i.e., when available input power is the maximum possible, basically to ensure the mechanical integrity of the bearings. At this wind velocity the achieved angular velocity is 32.87 krpm and the harvester bearing is ensured until 35 krpm of angular velocity.

The selection of the diode, MOSFET and architecture was performed and verified with different simulation tools, LTspice® [44] and (Power Electronics Simulation) PSIM® [45], however other simulation packages could be used. PSIM is mainly used for modelling basic architectures and LTspice for structural and behavioral models.

*4.1. Operation Principles of AC/DC Converters Architectures*

The operation of the architectures of the AC/DC converters used in this research work are explained briefly in the following lines.

4.1.1. Three Phase Diode Bridge

The three-phase diode bridge rectifier is taken as reference for the other topologies/architectures analyzed in this work. Figure 21 shows the architecture of three-phase diode bridge rectifier. The average DC output is improved using a capacitor between the diodes and resistive load, $C_{out}$. At the same time, this capacitor not only reduces the ripple of the signal but also acts as a storage device.

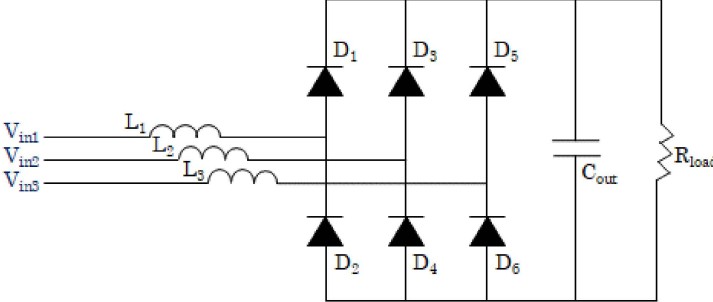

**Figure 21.** Three phase diode bridge architecture.

The waveforms present in a three-phase diode rectifier architecture are shown in Figure 22.

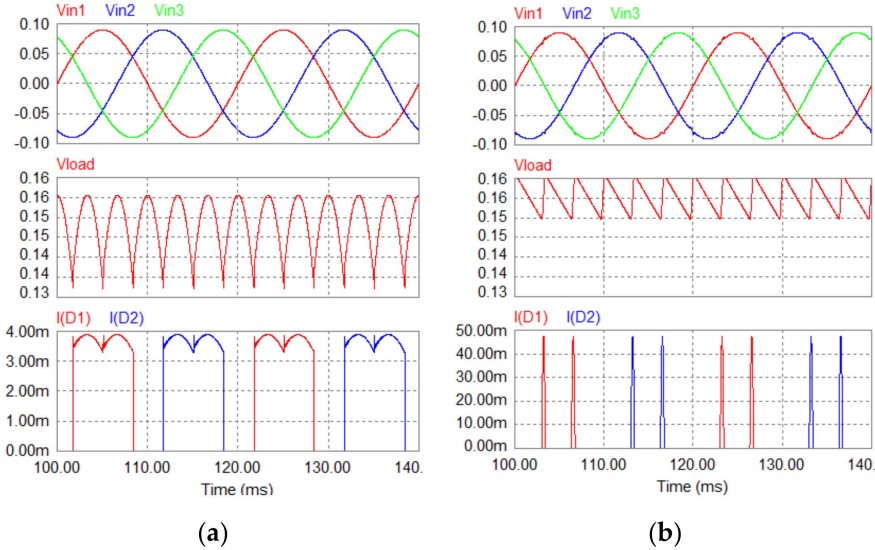

(**a**)                      (**b**)

**Figure 22.** Three-phase diode bridge waveforms obtained with PSIM. (**a**) With a low value output capacitance (10 nF). (**b**) With a high value output capacitance (68 µF).

Each of the three phases have the same behavior, with conduction angles for each diode of 120°, as the diode current waveforms of Figure 22a show. Focusing on the diode pair D1 and D2 of phase 1 (see the details of Figure 22a), D1 starts its conduction angle when its anode voltage, Vin1, is higher than its cathode voltage, Vin2 + Vin3. When D1 is in conduction, as D2 is off for its cathode voltage is higher than its anode one, the return path for the current could only be done through D4 for the first 60° of conduction and later through D6 for the remaining 60°. The ripple of the diode current and the output voltage waveforms are primarily functions of the load, capacitance value and the windings inductance. As the inductance is a fixed value, only the output capacitance value and the load change. Figure 22b shows the waveforms of the three-phase diode bridge when the output capacitance is high; the diodes conduction angles are reduced because the output capacitance has changed and provides most of the energy that the load demands, reducing the output voltage ripple, but increasing the peak current accordingly.

### 4.1.2. Secondary Side Diode-Based Topology

The circuit schematic of the secondary side diode-based [46] converter is Figure 23.
Figure 24 shows a LTspice simulation of circuit internal signals with 3 m/s and 10 m/s wind velocities.

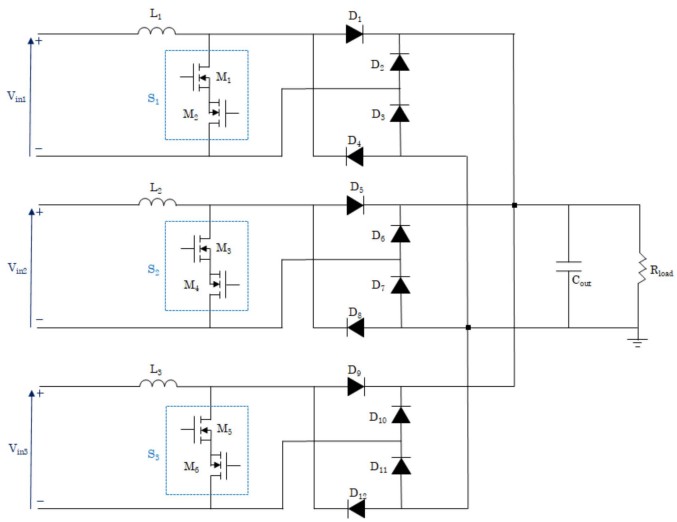

**Figure 23.** Secondary side diode three-phase circuit.

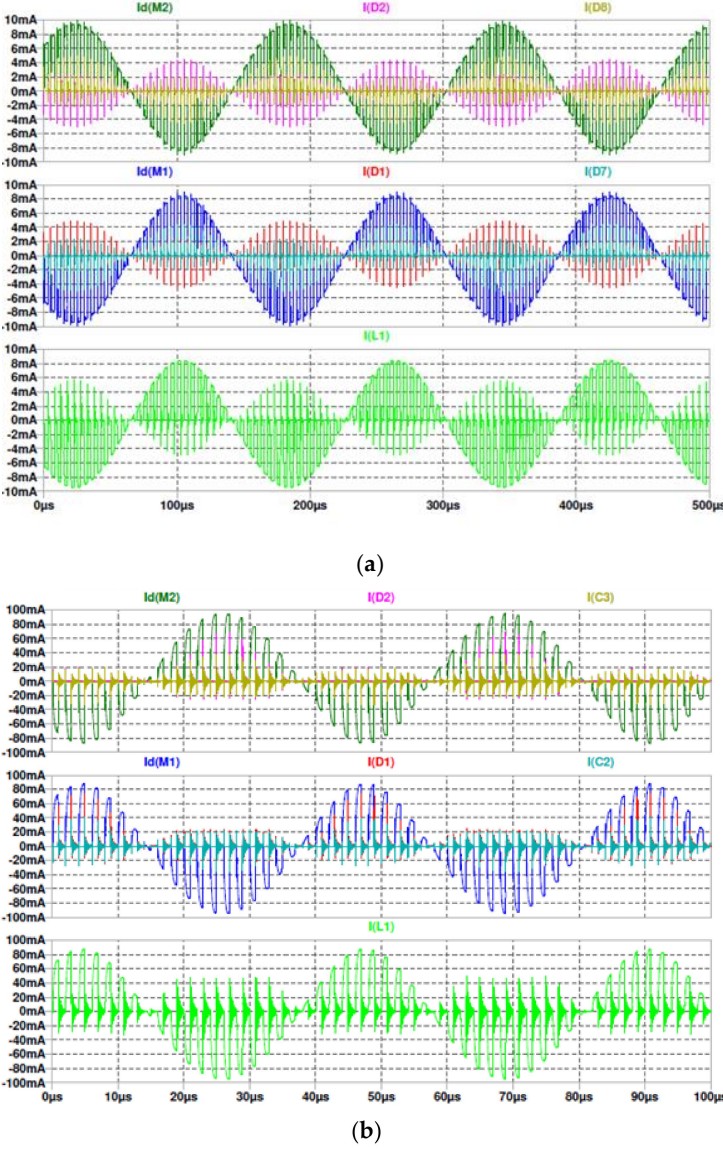

**Figure 24.** Circuit internal signals inside the architecture with 3 m/s wind velocity obtained with LTspice. (**a**) 3 m/s wind velocity. (**b**) 10 m/s wind velocity.

### 4.1.3. Split-N and Split-NP Topologies

Both topologies constitute an improvement of the previous one because Split-N and split-NP, unlike the previous of the rotational parts architectures, work like a switched power converter. The MOSFET control signals are generated with a pulse width modulator (PWM) controller that uses the measurement of the output voltage level to stabilize the output voltage, thus regulating the output voltage to the required value. The split capacitor topology-based converter [47,48] is composed of six MOSFETs ($M_1$, $M_2$, $M_3$, $M_4$, $M_5$, and $M_6$). They are connected in the same way as in the side-diode topology and allow the bidirectional switching. The diodes ($D_1$, $D_2$, $D_3$, $D_4$, $D_5$, and $D_6$) provide a rectifying path in each half cycle for each phase. Six capacitors (two per phase) with same capacitance value are used for charge recycling and to replace the diodes $D_3$, $D_4$, $D_7$, $D_8$, $D_{11}$ and $D_{12}$ of the side diode topology (Figure 21). Thanks to this, the diode forward voltage drop is avoided, which leads to have fewer losses and improves the system efficiency. Moreover, the output capacitor provides the required DC voltage for the resistive load. Within this type of architecture, two variations will be simulated and analyzed.

- Split topology with N-type MOSFETs
- Split topology with N and P-type MOSFETs

Split Topology with N-Type MOSFETs

So called N-type Split topology is composed of six N-type MOSFETs ($M_1$, $M_2$, $M_3$, $M_4$, $M_5$ and $M_6$). Compared with side-diode architecture, replacing diodes with capacitors in the rectification part of the circuit reduces the forward diode losses. This topology schematic is shown in Figure 25.

Analyzing a single phase, the split N architecture works similarly to the previous architecture, a step-up or boost switched converter, with two switches instead of one; thus, it is able to operate in both the positive and negative cycles of the AC input voltage. The capacitive divider C1 and C2 provides the reference voltage allowing the correct biasing of the switches, M1 and M2, and reducing the output voltage ripple alongside the output capacitor. The higher this value, the lower the voltage ripple.

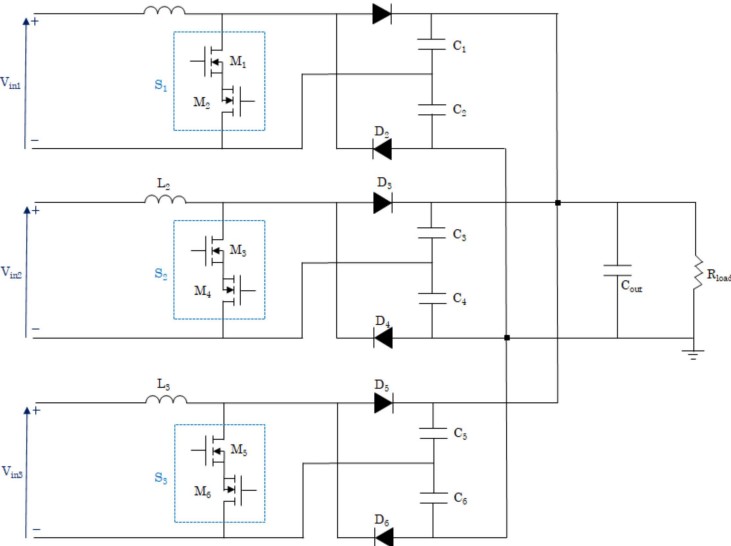

**Figure 25.** Split N capacitor three-phase circuit.

The waveforms this topology presents for a continuous mode operation of the boost converter are shown in Figure 26a. The switches M1 and M2 are continuously turning on and off with the clock frequency of the switched converter, $f_{sw}$, and a duty cycle, D, corresponding to the output voltage specified, $V_{load}$. The average output voltage could be estimated with the following expression (22),

which differs slightly from that of a DC-DC boost converter because the input waveform is AC, thus it has an extra term function of the switching frequency and its duty cycle, and the AC frequency:

$$V_{load} = \frac{V_{in1}}{(1-D)} \cdot \frac{D \cdot f_{sw}}{f_{AC}} \tag{22}$$

Thus, the average current through the inductance is (23):

$$I_L = \frac{V_{load}}{R_{load} \cdot (1-D)} \tag{23}$$

In addition, the maximum ripple of the inductance current is obtained through (24):

$$\Delta i_L(t)_{max} = \frac{V_{load} \cdot D}{f \cdot L_1} \tag{24}$$

where $D$ is the duty cycle of the converter, $V_{in1}$ is the root mean square (rms) value of the AC voltage of a single phase, $f_{AC}$ is the frequency of the AC signal, $L_1$ is the inductance of one winding and $f$ is the switching frequency of the converter.

On the positive cycle of the $V_{in}$ signal, when the switches are on, the current flows from M1 to M2 back to the source and, when M1 and M2 are in the off state, the current flows to the load through D1, Figure 26b. On the negative cycle of $V_{in}$, the current flow changes polarity, flowing from M2 to M1, in their conduction states. When the switches turn off, the current flows to the load through D2. Therefore, the current through the load always has the same polarity.

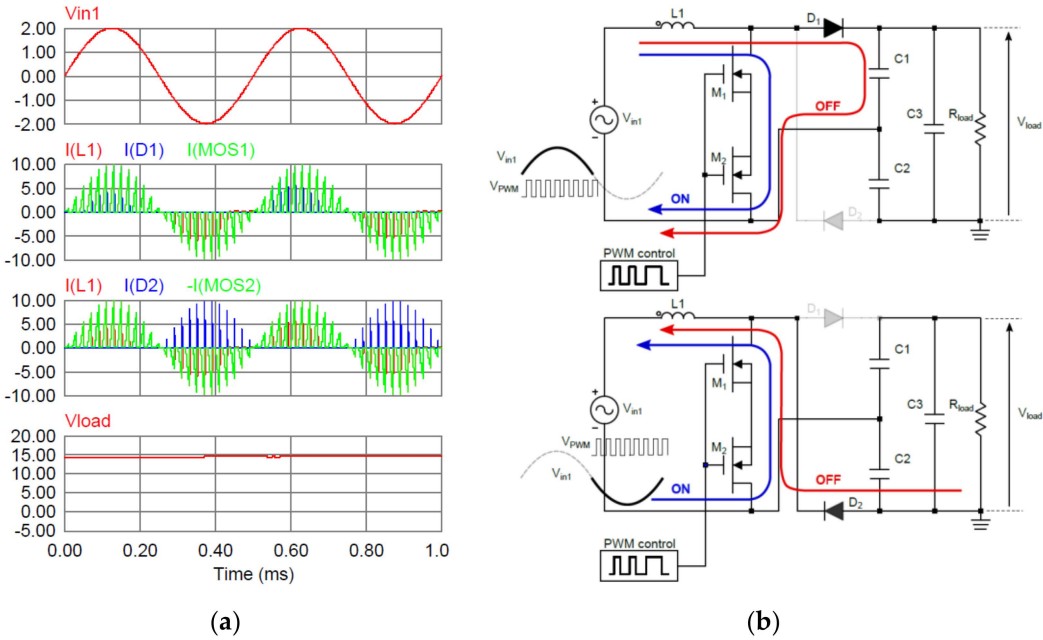

**Figure 26.** Waveforms and states of a Split N converter obtained with PSIM. (**a**) Current waveforms and output load voltage. (**b**) States of devices and current flows.

Split Topology with N and P-Type MOSFETs

The denominated Split NP has the same behavior as the Split-N, the difference is that the topology is composed of six MOSFETs, three of P-type ($M_1$, $M_3$, and $M_5$) and another three of N-type ($M_2$, $M_4$, and $M_6$). Therefore, the PWM control signal applied to the P-type MOSFET must becomplementary of that of the N-type MOSFET. This topology schematic is shown in Figure 27.

One simulation of circuit internal signals with 3 and 10 m/s wind velocities is shown in Figure 28.

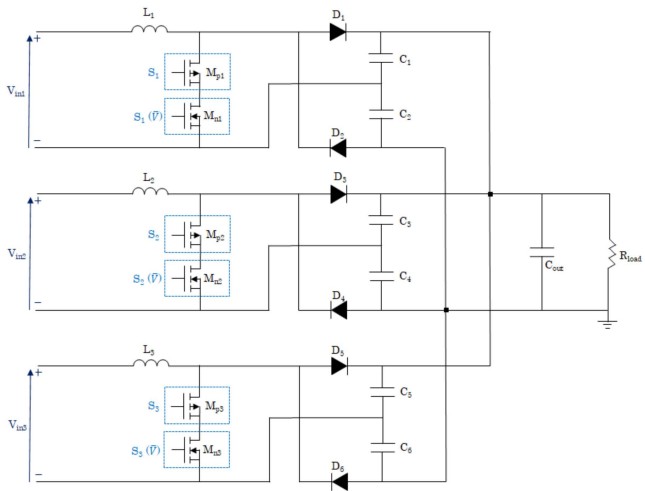

**Figure 27.** Split capacitor three-phase circuit with P and N MOSFETs, Split NP.

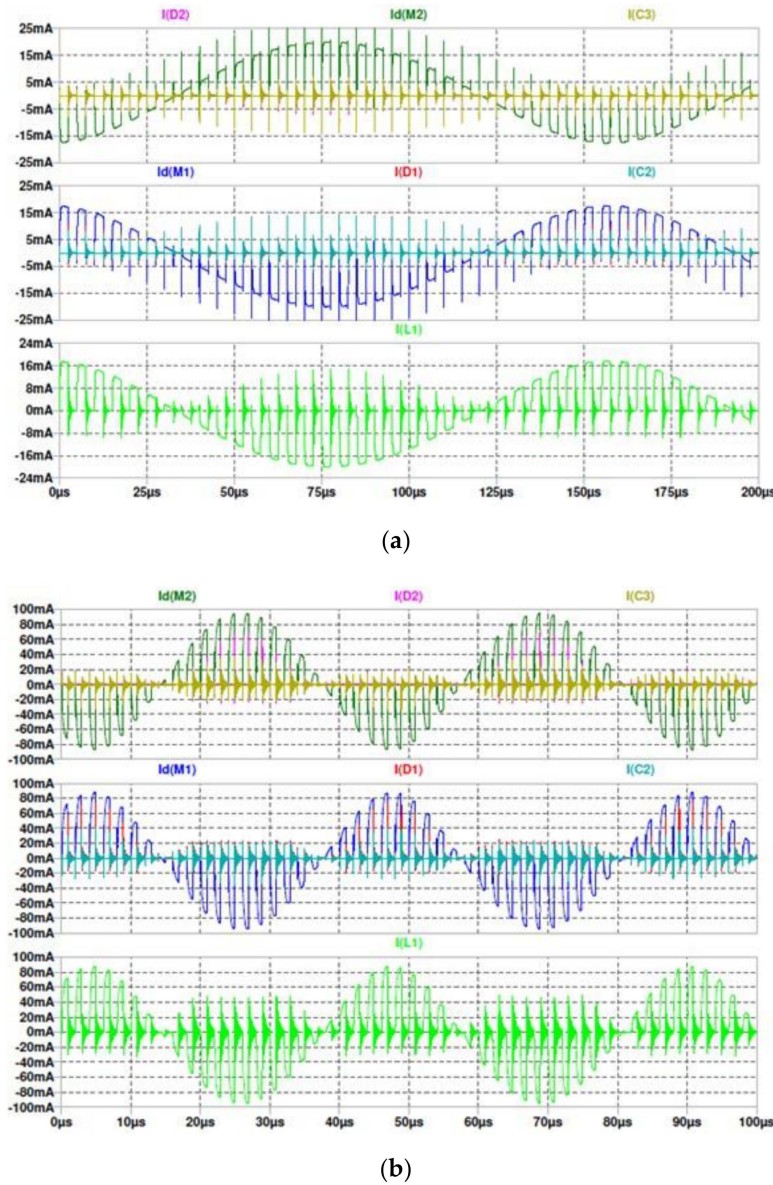

**Figure 28.** Circuit internal signals inside the architecture. (**a**) A 3 m/s wind velocity. (**b**) A 10 m/s wind velocity.

### 4.2. Diode Selection

The diode selection is done reviewing several commercial diodes based on their characteristics and aiming for low conductions and switching losses. The diode bridge architecture, shown in Figure 21, was simulated and tested for the minimum and maximum wind velocities achieved with the harvester prototype, using the considered diode's Spice model. In Tables 17 and 18, and Figure 29, the results achieved in the specified conditions are shown.

**Table 17.** Comparative results between different diodes at 3 m/s.

| Diode | $R_{load}$ (Ω) | $f_{in}$ (kHz) | $V_{in}$ (Vp) | $V_{out}$ (V) | $I_{out}$ (mA) | $P_{out}$ (mW) | $P_{in}$ (mW) | $\eta$ (%) |
|---|---|---|---|---|---|---|---|---|
| MBR0520L [49] | 40 | 1.768 | 0.886 | 0.405 | 10.147 | 4.118 | 7.359 | 55.96 |
| BAT15-03W [50] | 40 | 1.944 | 1.209 | 0.309 | 7.73 | 2.39 | 13.703 | 17.44 |
| 1PS79SB31 [51] | 40 | 1.808 | 0.948 | 0.391 | 9.782 | 3.827 | 8.425 | 45.42 |
| NSR20F30NXT5G [52] | 40 | 1.782 | 0.904 | 0.398 | 9.974 | 3.979 | 7.661 | 51.94 |
| NSVR351SDSA3 [53] | 40 | 1.847 | 1.019 | 0.374 | 9.353 | 3.499 | 9.734 | 35.94 |
| STPS20L15G [54] | 40 | 1.546 | 0.584 | 0.394 | 13.154 | 5.191 | 12.789 | 40.59 |

**Table 18.** Comparative results between different diodes at 10 m/s.

| Diode | $R_{load}$ (Ω) | $f_{in}$ (kHz) | $V_{in}$ (Vp) | $V_{out}$ (V) | $I_{out}$ (mA) | $P_{out}$ (mW) | $P_{in}$ (mW) | $\eta$ (%) |
|---|---|---|---|---|---|---|---|---|
| MBR0520L | 30 | 6.717 | 2.575 | 1.995 | 66.572 | 132.957 | 248.648 | 53.47 |
| BAT15-03W | 30 | 7.629 | 3.9 | 1.753 | 58.451 | 102.497 | 190.125 | 53.91 |
| 1PS79SB31 | 30 | 6.857 | 2.751 | 1.972 | 65.649 | 129.646 | 283.80 | 45.68 |
| NSR20F30NXT5G | 30 | 6.73 | 2.588 | 1.992 | 66.405 | 132.291 | 251.165 | 52.67 |
| NSVR351SDSA3 | 30 | 7.106 | 3.1 | 1.915 | 63.961 | 122.73 | 360.375 | 34.06 |
| STPS20L15G | 30 | 6.497 | 2.321 | 2.047 | 68.261 | 139.789 | 202.014 | 69.20 |

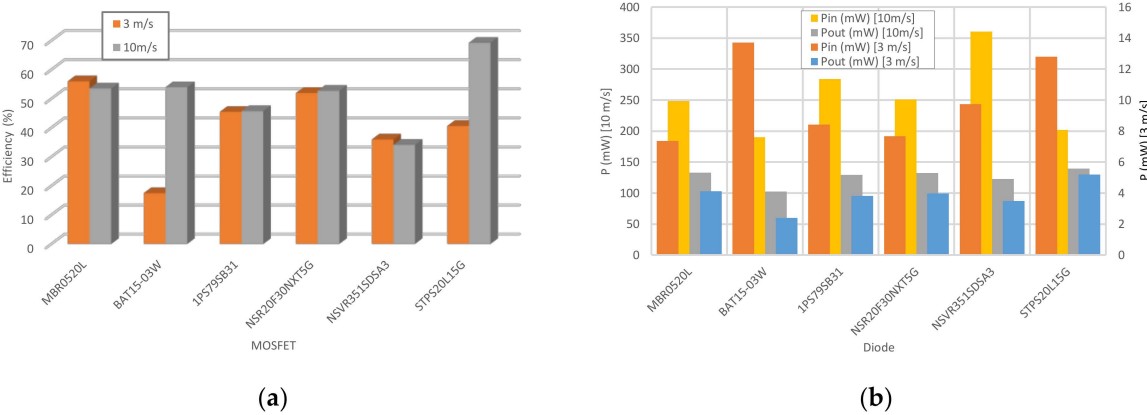

(**a**)                                  (**b**)

**Figure 29.** Comparisons between diodes at 3 m/s and 10 m/s. (**a**) Efficiency. (**b**) Input/output power.

In low wind velocity conditions, with MBR0520L, the efficiency reaches 55.96% and, with STPS20L15G in high conditions, 69.20%. However, in high wind velocity conditions, STPS20L15G achieves an efficiency of 40.59%, meanwhile MBR0520L achieves 53.47% in low conditions. Thus, due to the stability that MBR0520L provides in both conditions, this is the chosen diode.

### 4.3. MOSFET Selection

The MOSFET selection is carried out with the same procedure as with diodes. However, in this case, simulations were made with Split N capacitor topology (Figure 25).

The selection of the MOSFETs was done in the following conditions: at minimum input state (3 m/s wind velocity): 500 kHz switching frequency, 0.05/0.1 duty-cycle (D), $R_{load}$ = 80/90 Ω and the commercial diode model MBR0520L; and at maximum input state (10m/s): 1000 kHz switching frequency, $R_{load}$ = 70/80 Ω and maintaining duty-cycle values. With these test conditions, several commercial MOSFETs were selected based on their appropriate characteristics for the present use, i.e.,

minimizing the losses. In Tables 19 and 20 and Figure 30, the achieved results with detailed conditions are shown.

**Table 19.** Comparative results between different MOSFETs at 3 m/s.

| MOSFET | $V_{in}$ (Vp) | $f_{in}$ (kHz) | $R_{load}$ (Ω) | D | $f_{sw}$ (kHz) | $Q_g$ (nC) | $R_{ds}$ (mΩ) | $V_{out}$ (V) | $I_{out}$ (mA) | $P_{out}$ (mW) | $P_{contr.}$ (mW) | $P_{out}$ (mW) | $\eta$ (%) |
|---|---|---|---|---|---|---|---|---|---|---|---|---|---|
| Si1555DL [55] | 1.025 | 1.9 | 80 | 0.05 | 500 | 0.8 | 0.63 | 0.677 | 8.469 | 5.73 | 0.523 | 5.20 | 44.05 |
| Si1555DL | 1.025 | 1.9 | 90 | 0.1 | 500 | 0.8 | 0.63 | 0.709 | 7.887 | 5.59 | 0.527 | 5.063 | 42.83 |
| Si8466EDB [56] | 1.025 | 1.9 | 80 | 0.05 | 500 | 6.8 | 0.05 | 0.683 | 8.538 | 5.83 | 6.79 | −0.96 | −8.12 |
| Si8466EDB | 1.025 | 1.9 | 90 | 0.1 | 500 | 6.8 | 0.05 | 0.719 | 7.997 | 5.74 | 7.313 | −1.57 | −13.30 |
| IRLU3802 [57] | 1.025 | 1.9 | 80 | 0.05 | 500 | 27 | 8.5 | 0.664 | 8.303 | 5.51 | 20.476 | −14.96 | −126.6 |
| IRLU3802 | 1.025 | 1.9 | 80 | 0.1 | 500 | 27 | 8.5 | 0.657 | 8.224 | 5.40 | 20.4 | −15 | −126.9 |
| Si1553CDLN [58] | 1.025 | 1.9 | 80 | 0.05 | 500 | 0.55 | 0.578 | 0.68 | 8.51 | 5.78 | 0.341 | 5.43 | 46.01 |
| Si1553CDLN | 1.025 | 1.9 | 90 | 0.1 | 500 | 0.55 | 0.578 | 0.715 | 7.945 | 5.68 | 0.350 | 5.33 | 45.09 |

**Table 20.** Comparative results between different MOSFETs at 10 m/s.

| MOSFET | $V_{in}$ (Vp) | $f_{in}$ (kHz) | $R_{load}$ (Ω) | D | $f_{sw}$ (kHz) | $Q_g$ (nC) | $R_{ds}$ (mΩ) | $V_{out}$ (V) | $I_{out}$ (mA) | $P_{out}$ (mW) | $P_{contr.}$ (mW) | $P_{out}$ (mW) | $\eta$ (%) |
|---|---|---|---|---|---|---|---|---|---|---|---|---|---|
| Si1555DL | 4.17 | 8.766 | 70 | 0.05 | 1000 | 0.8 | 0.63 | 3.252 | 46.471 | 151.12 | 1.027 | 150.09 | 77 |
| Si1555DL | 4.17 | 8.766 | 80 | 0.1 | 1000 | 0.8 | 0.63 | 3.451 | 43.147 | 148.90 | 1.1 | 147.83 | 75.90 |
| Si8466EDB | 4.17 | 8.766 | 70 | 0.05 | 1000 | 6.8 | 0.05 | 3.252 | 46.47 | 151.12 | 12.374 | 138.74 | 71.23 |
| Si8466EDB | 4.17 | 8.766 | 80 | 0.1 | 1000 | 6.8 | 0.05 | 3.451 | 43.146 | 148.89 | 14.505 | 134.38 | 69 |
| IRLU3802 | 4.17 | 8.766 | 70 | 0.05 | 1000 | 27 | 8.5 | 3.2 | 45.724 | 146.31 | 36.684 | 109.62 | 56.28 |
| IRLU3802 | 4.17 | 8.766 | 80 | 0.1 | 1000 | 27 | 8.5 | 3.398 | 42.486 | 144.36 | 37.422 | 106.938 | 54.90 |
| Si1553CDLN | 4.17 | 8.766 | 70 | 0.05 | 1000 | 0.55 | 0.578 | 3.264 | 46.638 | 152.22 | 0.519 | 151.70 | 77.89 |
| Si1553CDLN | 4.17 | 8.766 | 80 | 0.1 | 1000 | 0.55 | 0.578 | 3.471 | 43.389 | 150.60 | 0.57 | 150 | 77.033 |

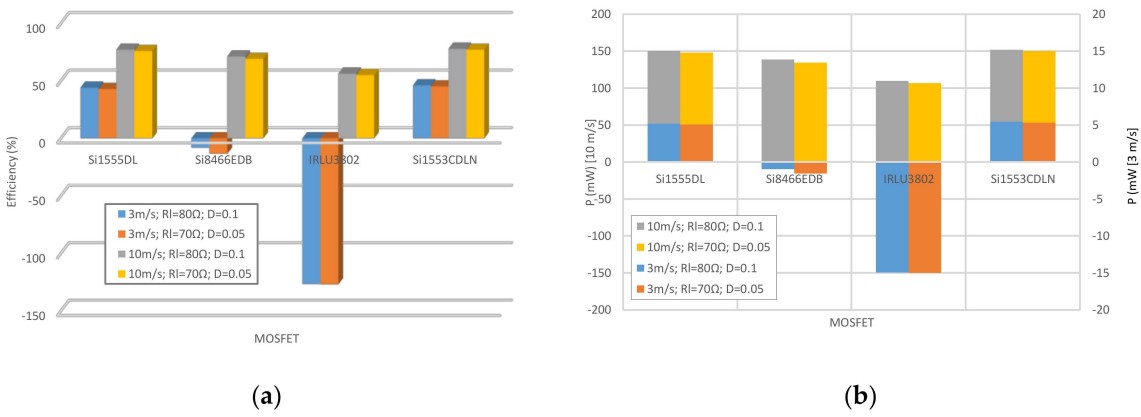

**Figure 30.** Comparisons between diodes at 3 m/s and 10m/s. (**a**) Efficiency. (**b**) Output power.

For the minimum wind velocity, the Si1553CDLN MOSFET is the one that provides the maximum efficiency of 46.01%. Moreover, there is a couple of high losses devices. These are Si8466EDB and IRLU3802, due to the low value of the income power. The high losses figure with the IRLU3802 MOSFET occurs because its consumption is higher than energy harvested by the system. When the income power is at its maximum, the efficiencies of all the devices are positive and range between 55% and 78%. In these conditions, the Si1553CDLN MOSFET grants to the converter a maximum efficiency of 77.8%. Moreover, Si1553CDLN has the lowest consumption in both work operating modes, providing the converter its maximum efficiency and output power. Therefore, Si1553CDLN MOSFET was selected for the AC/DC converter. A close look at the Si1553CDLN characteristics parameter of Tables 18 and 19 shows that it is the one with best relationship between Ron resistance and gate capacitance, thus, achieving lower switching and conduction losses.

### 4.4. Architectures: Evaluation and Selection

The selection of an AC/DC architecture aims to obtain the maximum efficiency. Four architectures were analyzed in order to achieve this goal. Although rectifiers were discarded previously, the first one

is rectifier diode based, which is used as reference and for comparison. The other three alternatives are AC/DC converter topologies that rectify and boost the input signal.

### 4.4.1. Three-Phase Diode Bridge Rectifier

The results of the simulations for the rested wind velocity conditions are shown Tables 21 and 22 and Figure 31.

**Table 21.** Diode bridge simulation results under the minimum input condition.

| $R_{load}$ ($\Omega$) | $V_{in}$ (V) | $f_{in}$ (kHz) | $V_{out}$ (V) | $I_{out}$ (mA) | $P_{out}$ (mW) | $\eta_{generator}$ (%) | $\eta_{converter}$ (%) | $\eta_{general}$ (%) |
|---|---|---|---|---|---|---|---|---|
| 10 | 0.66 | 1.687 | 0.17 | 17.31 | 2.97 | 16.75 | 27.13 | 4.54 |
| 20 | 0.76 | 1.735 | 0.27 | 13.83 | 3.81 | 16.43 | 34.24 | 5.63 |
| 30 | 0.83 | 1.756 | 0.34 | 11.67 | 4.04 | 15.83 | 36.76 | 5.82 |
| 40 | 0.88 | 1.768 | 0.40 | 10 | 4.04 | 15.17 | 37.69 | 5.72 |
| 50 | 0.92 | 1.776 | 0.44 | 8.94 | 4 | 14.57 | 37.78 | 5.50 |
| 60 | 0.96 | 1.781 | 0.48 | 8.02 | 3.85 | 14.02 | 37.49 | 5.26 |
| 70 | 0.99 | 1.785 | 0.50 | 7.25 | 3.67 | 13.53 | 37 | 5.01 |
| 80 | 1.01 | 1.788 | 0.53 | 6.65 | 3.54 | 13.09 | 36.4 | 4.76 |
| 90 | 1.03 | 1.79 | 0.55 | 6.13 | 3.38 | 12.71 | 35.7 | 4.54 |
| 100 | 1.05 | 1.792 | 0.57 | 5.71 | 3.26 | 12.35 | 35.02 | 4.32 |

**Table 22.** Diode bridge simulation results under the maximum input condition.

| $R_{load}$ ($\Omega$) | $V_{in}$ (V) | $f_{in}$ (kHz) | $V_{out}$ (V) | $I_{out}$ (mA) | $P_{out}$ (mW) | $\eta_{generator}$ (%) | $\eta_{converter}$ (%) | $\eta_{general}$ (%) |
|---|---|---|---|---|---|---|---|---|
| 10 | 1.64 | 6.63 | 1.3 | 1.32 | 113.02 | 10.34 | 58.94 | 6.09 |
| 20 | 2.20 | 6.694 | 1.62 | 81.12 | 131.58 | 11.19 | 60.78 | 6.80 |
| 30 | 2.57 | 6.717 | 1.99 | 66.42 | 132.32 | 11.23 | 58.79 | 6.60 |
| 40 | 2.84 | 6.729 | 2.26 | 56.55 | 127.93 | 11.06 | 56.12 | 6.21 |
| 50 | 3.04 | 6.736 | 2.47 | 49.40 | 122.02 | 10.82 | 53.44 | 5.78 |
| 60 | 3.21 | 6.742 | 2.63 | 43.90 | 115.65 | 10.57 | 50.94 | 5.38 |
| 70 | 3.34 | 6.745 | 2.77 | 39.6 | 109.78 | 10.33 | 48.62 | 5.02 |
| 80 | 3.45 | 6.748 | 2.88 | 36.02 | 103.78 | 10.11 | 46.45 | 4.70 |
| 90 | 3.54 | 6.75 | 2.97 | 33.08 | 98.50 | 9.91 | 44.44 | 4.40 |
| 100 | 3.63 | 6.752 | 3.1 | 30.61 | 93.70 | 9.72 | 42.64 | 4.15 |

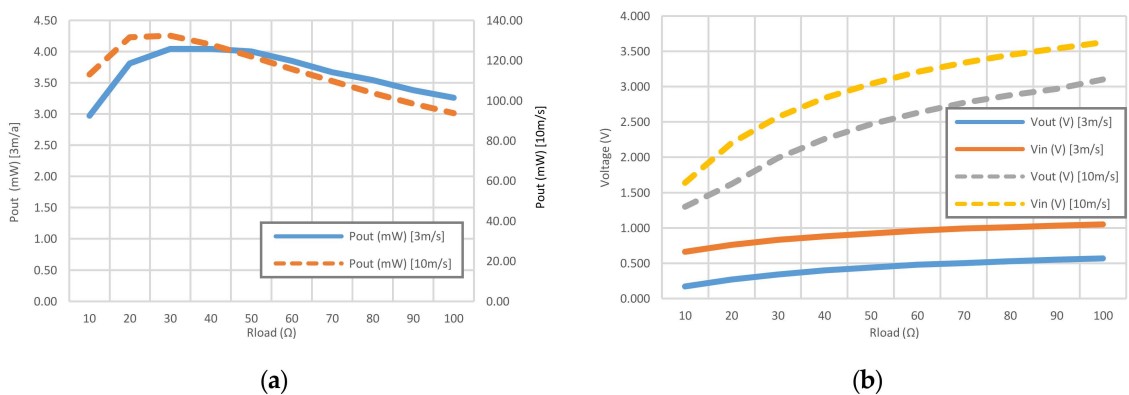

(**a**)                    (**b**)

**Figure 31.** *Cont.*

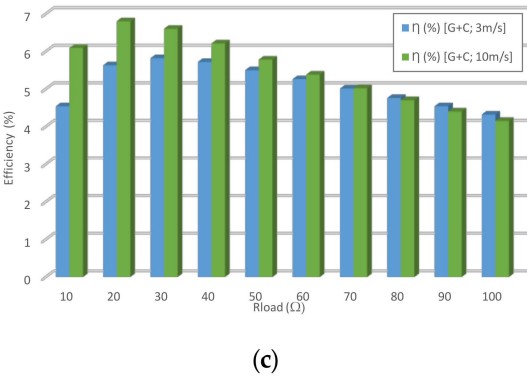

(**c**)

**Figure 31.** Output power, voltage and efficiency values with different resistive loads at 3 m/s and 10 m/s wind velocities. (**a**) Pout (mW). (**b**) Voltage (V). (**c**) Efficiency of the system (%): Generator + Conversion system.

For the test case of minimum wind velocity, the maximum output power is 4.04 mW, and it is achieved for resistive loads of 30 Ω or 40 Ω. The maximum voltage level obtained is 0.57 V for a 100 Ω load and 3.26 mW is obtained at the output. The required resistive load value to achieve a 3 V level would be so big that the output power would be close to zero, i.e., an open circuit condition. Thus, this architecture is not valid. Nevertheless, simulations were carried out for comparison between architectures.

For a wind velocity of 10 m/s, the output level of 3 V is only reached for a resistive load of 100 Ω, obtaining 93.70 mW. The maximum obtained power is 132.32 mW for 30 Ω load and 1.99 V signal.

Rectifier efficiencies are 38% and 61% for 3 m/s and 10 m/s wind velocities, respectively. The complete system efficiency is around 5% in both ambient conditions.

In both cases, the input voltage is lower than the output voltage. Diode bridge architecture only rectifies input signal and would require a second stage to step-up the voltage; consequently, it is not a feasible alternative.

### 4.4.2. Secondary Side Diode-Based Topology

In order to verify the maximum output power and efficiency of the architecture, simulations were performed changing the values of different variables, such as switching frequency, duty-cycle, and resistive load.

Figure 32 shows the simulation results (output power, output voltage, generator efficiency, converter efficiency and complete system efficiency) with different wind velocity. Sweeps were carried out with resistive loads between 100 Ω and 2100 Ω, duty-cycles between 0.1 and 0.9, and switching frequencies of 200 kHz and 333 kHz for 3 m/s wind velocity, and 500 kHz and 1000 kHz for 10 m/s wind velocity. The switching frequency depends on the ambient conditions because the input range of wind speed is wide, thus a shift of the frequency for each case is required. The maximum output power and system efficiencies are obtained by sweeping the variables in the simulation for different test conditions.

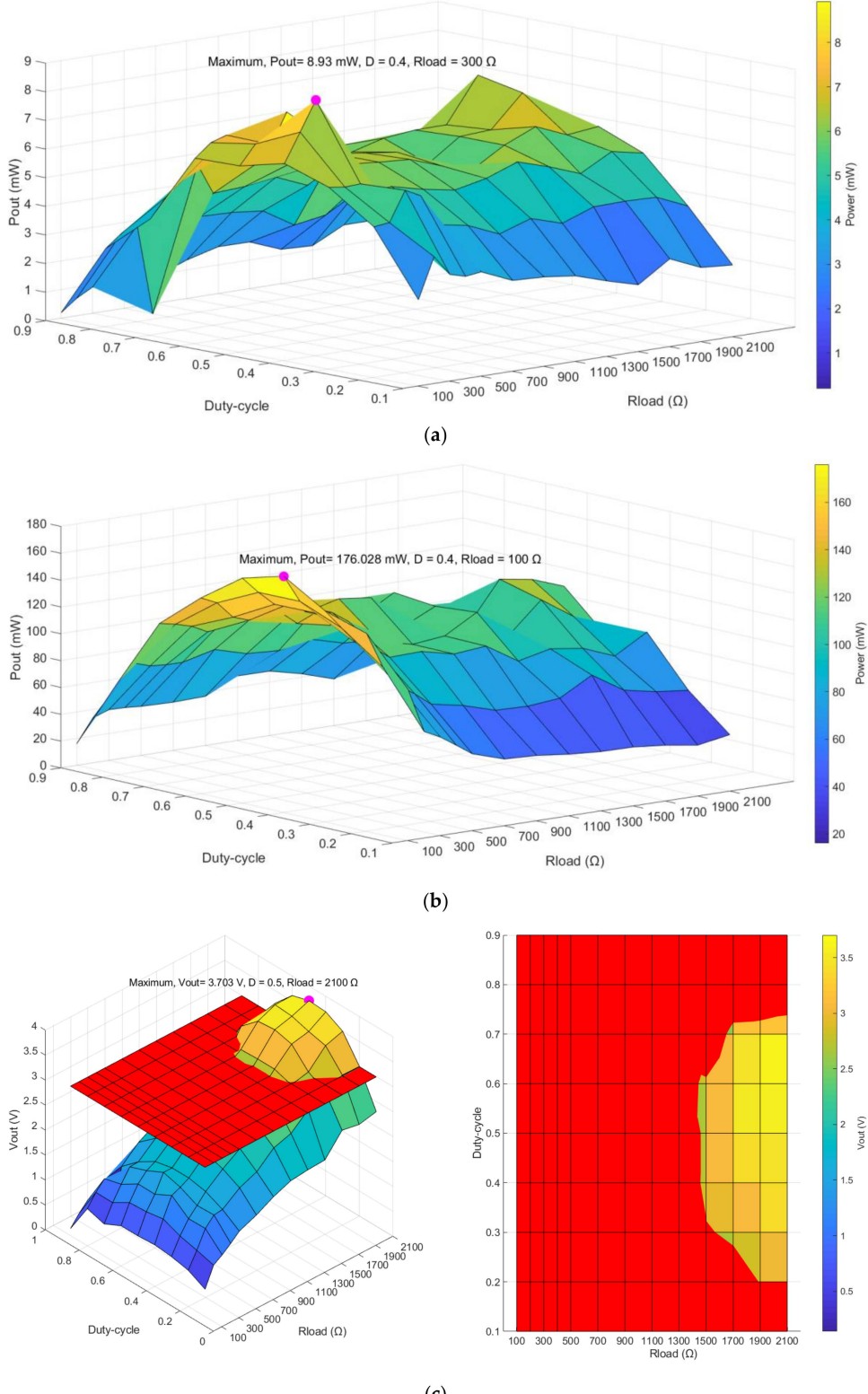

**Figure 32.** *Cont.*

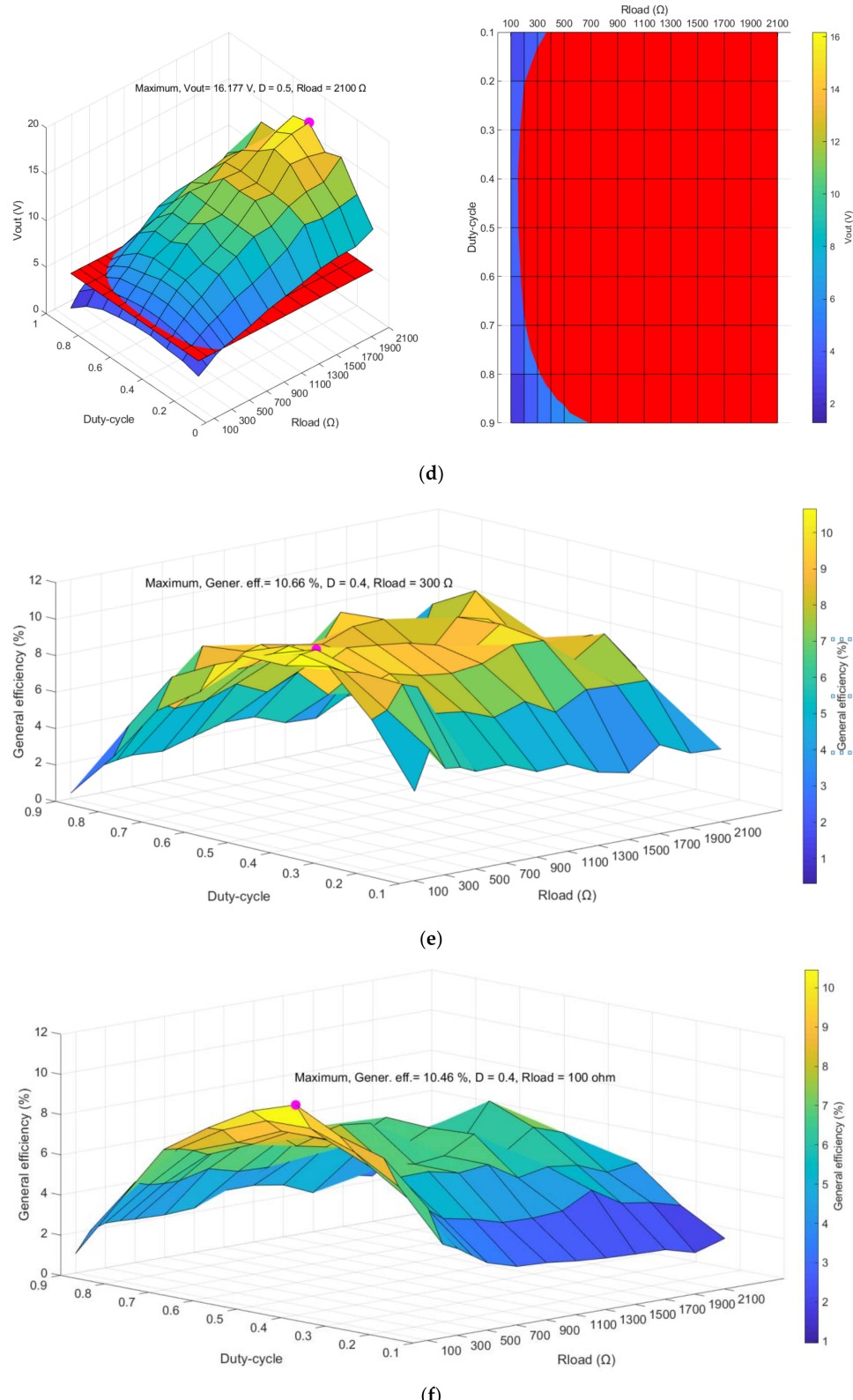

**Figure 32.** Three phase diode bridge converter simulation results. The (**a**,**c**,**e**) figures are with 3 m/s wind velocity and 200 kHz switching frequency. The (**b**,**d**,**f**) figures are with 10 m/s wind velocity and 500 kHz switching frequency. (**a**) Output power (3 m/s, 200 kHz). (**b**) Output power (10 m/s; 500 kHz). (**c**) Output voltage (3 m/s, 200 kHz), Red plane: 3 V as minimum. (**d**) Output voltage (10 m/s, 500 kHz), Red plane: 3 V as minimum. (**e**) Generator + Converter efficiency (3 m/s, 200 kHz). (**f**) Generator + Converter efficiency (10 m/s; 500 kHz).

For the lowest wind velocity (3 m/s) the maximum efficiency of the generator is 18.06% for a 100 Ω load and 0.3 duty-cycle. The efficiencies of the converter are higher than 40%, achieving a maximum efficiency of 81.09% with a duty-cycle of 0.6 for a 1100 Ω resistive load. Moreover, the complete system maximum efficiency is 10.66% for a 300 Ω load and 0.4 duty-cycle.

For the highest wind velocity (10 m/s) the maximum efficiency of the generator is 13.31% for a 1700 Ω load and 0.4 duty-cycle. The converter achieves a maximum efficiency of 81.94% with a duty-cycle of 0.2 for a 100 Ω resistive load. Moreover, the complete system maximum efficiency is 10.46% for a 100 Ω load and 0.4 duty-cycle.

### 4.4.3. Split Capacitor Topology

Split Topology with N-Type MOSFETs

Figure 33 shows simulation results (output power, output voltage, generator efficiency, converter efficiency and complete system efficiency) in function of resistive and duty-cycle, with the same switching frequency as in the previous case.

In the first case (3 m/s wind velocity), the maximum efficiency of the generator is 26.02% for a 300 Ω load and 0.4 duty-cycle. The converter efficiencies are higher than 40%, achieving maximum efficiency of 79.77% with a duty-cycle of 0.4 for a 700 Ω resistive load. The whole system maximum efficiency is 14% for a 100 Ω load and 0.4 duty-cycle.

In the second case (10 m/s), the maximum efficiency of the generator is 15.34% for a 1900 Ω load and 0.5 duty-cycle. The converter achieves a maximum efficiency of 79.06% with a duty-cycle of 0.5 for a 1100 Ω resistive load. The whole system maximum efficiency is 14% for 100 Ω load and 0.4 duty-cycle.

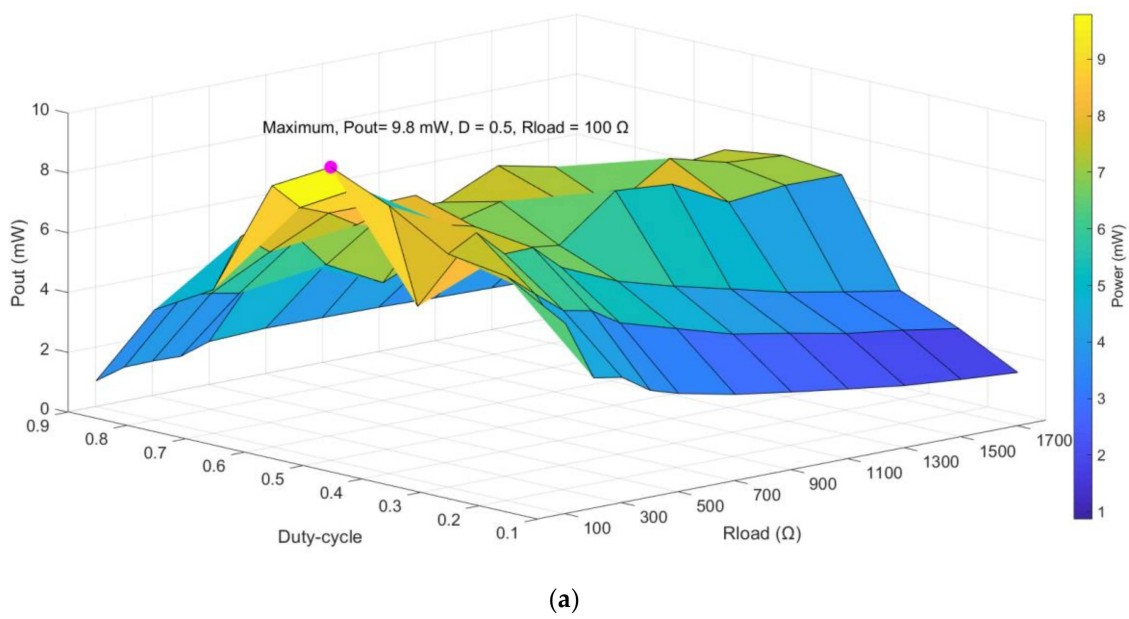

(**a**)

**Figure 33.** *Cont.*

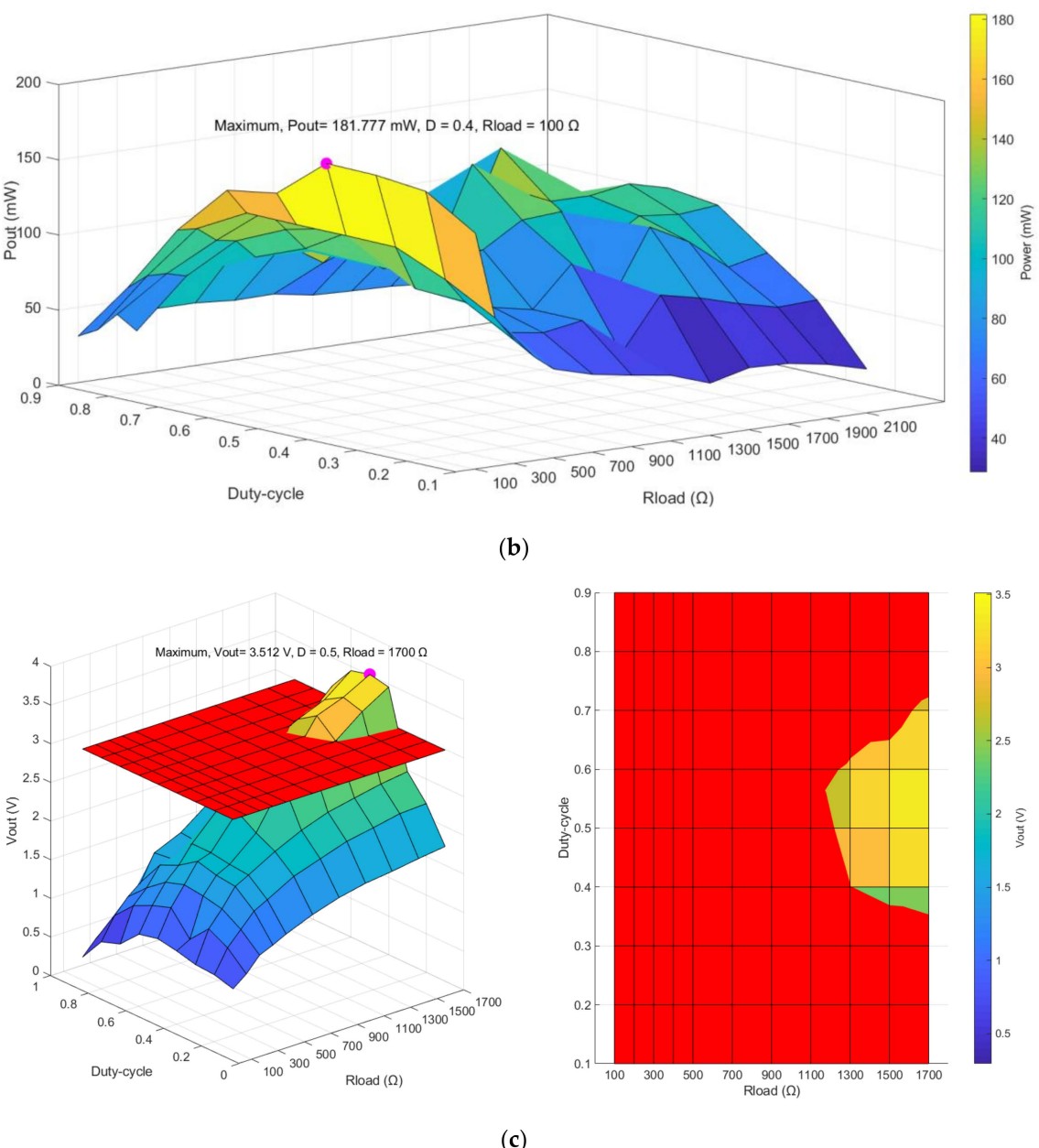

(**b**)

(**c**)

**Figure 33.** *Cont.*

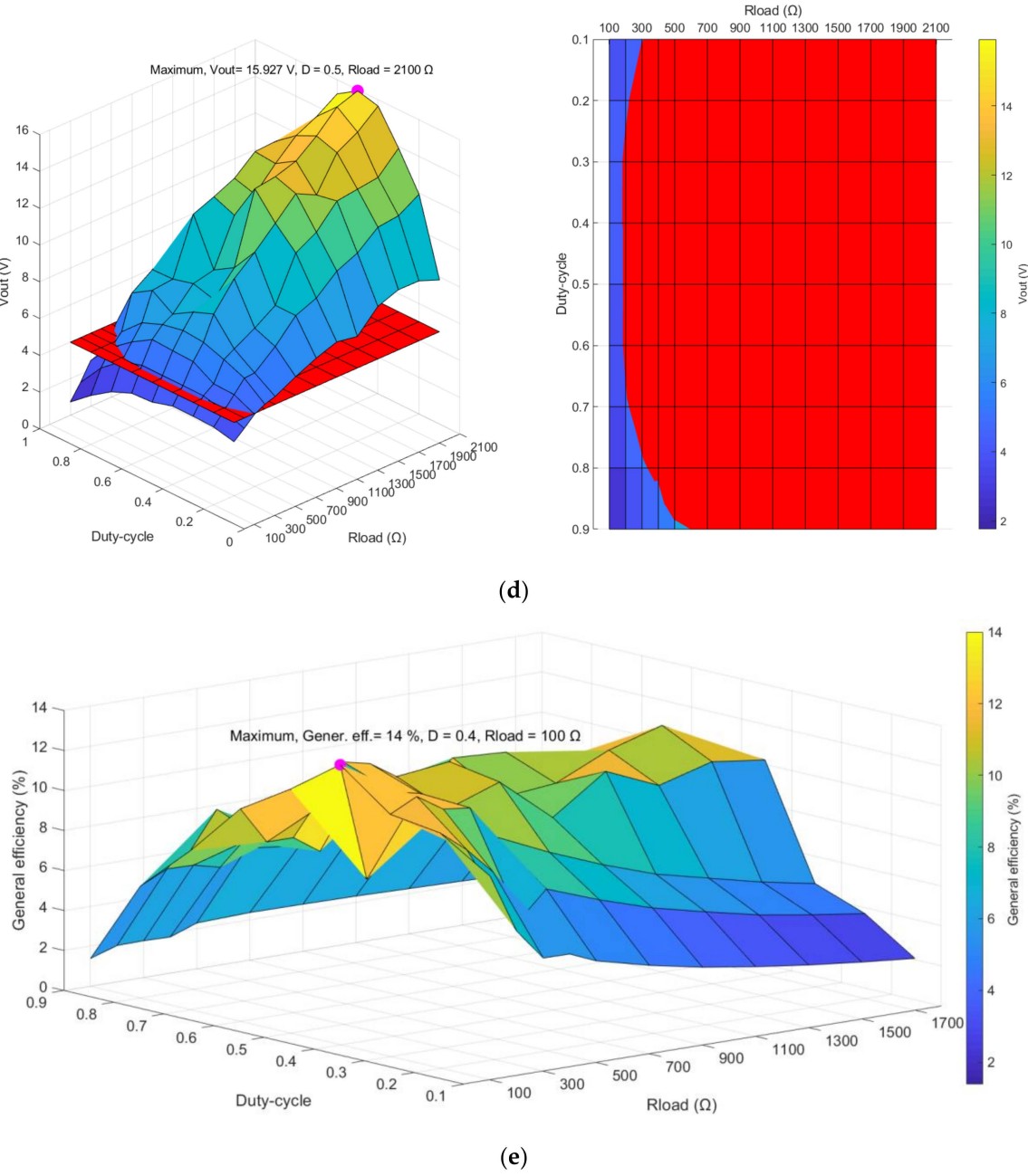

(**d**)

(**e**)

**Figure 33.** *Cont.*

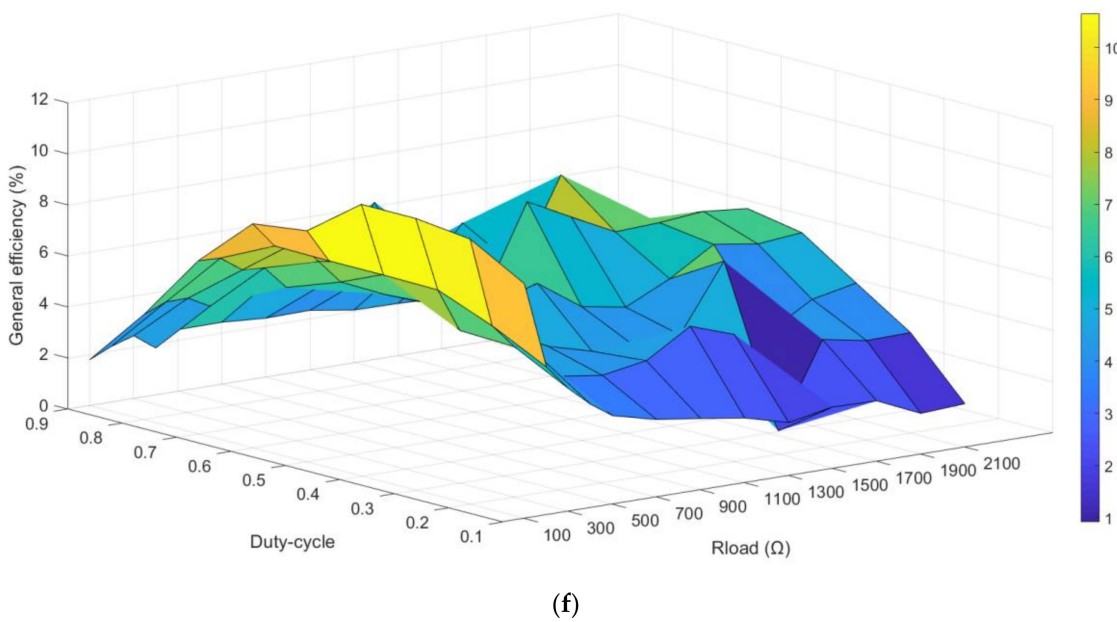

(**f**)

**Figure 33.** Split N converter simulation results. The (**a,c,e**) figures are with 3 m/s wind velocity and 333 kHz switching frequency. The (**b,d,f**) figures are with 10 m/s wind velocity and 500 kHz switching frequency. (**a**) Output power (3 m/s, 333 kHz). (**b**) Output power (10 m/s; 500 kHz). (**c**) Output voltage (3 m/s, 333 kHz), Red plane: 3 V as minimum. (**d**) Output voltage (10 m/s, 500 kHz), Red plane: 3 V as minimum. (**e**) Generator + Converter efficiency (3 m/s, 333 kHz). (**f**) Generator + Converter efficiency (10 m/s; 500 kHz).

Split Topology with N and P-Type MOSFETs

Figure 34 shows simulation results (output power, output voltage, generator efficiency, converter efficiency and complete system efficiency) as a function of resistive and duty-cycle, with the same switching frequency as in the previous case.

In the first case (3 m/s) the maximum efficiency of the generator is 23.78% for a 100 Ω load and 0.5 duty-cycle, being similar to 24% achieved with a 27 Ω load without any converter or rectifier, i.e., perfect impedance matching. The converter efficiencies are higher than 50%, achieving a maximum efficiency of 82.41% with a duty-cycle of 0.6 for a resistive load of 500 Ω. The whole system's maximum efficiency is 15.96% for a 100 Ω load and 0.5 duty-cycle.

In the second case (10 m/s wind velocity), the maximum efficiency of the generator is 14.87% for a 200 Ω load and 0.4 duty-cycle. The converter achieves a maximum efficiency of 76.66% with a duty-cycle of 0.1 for a 100 Ω resistive load. The whole system maximum efficiency is 11% for a 100 Ω load and 0.4 duty-cycle.

To conclude, analyzing all the sweeps of generator efficiency, it can be seen that the output power value at the resistive load is directly related to the best efficiency of the system. In addition, the overall efficiency depends more on the efficiency of the generator than on the efficiency of the converter. Consequently, the generator performance has more weight in the system efficiency.

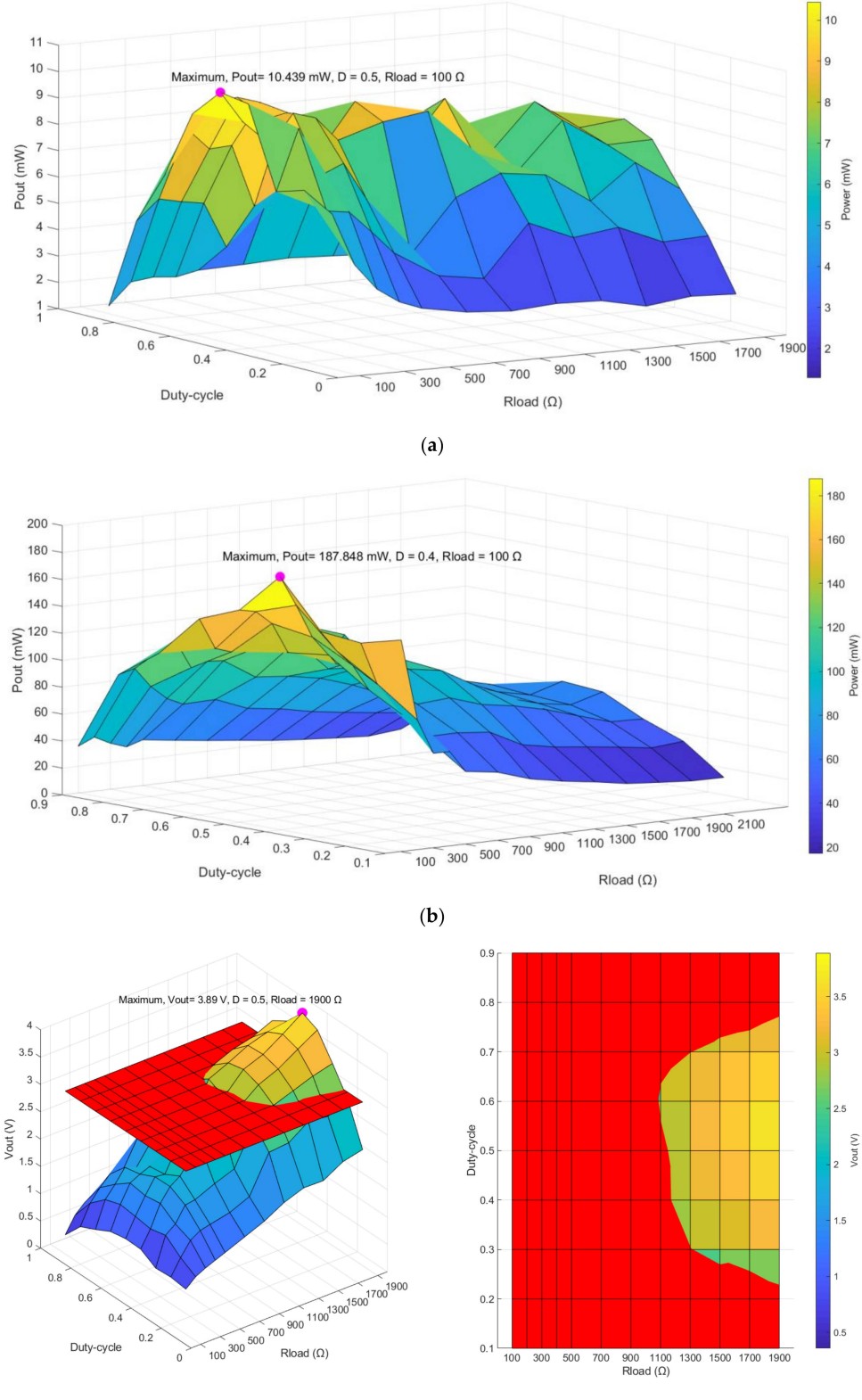

**Figure 34.** *Cont.*

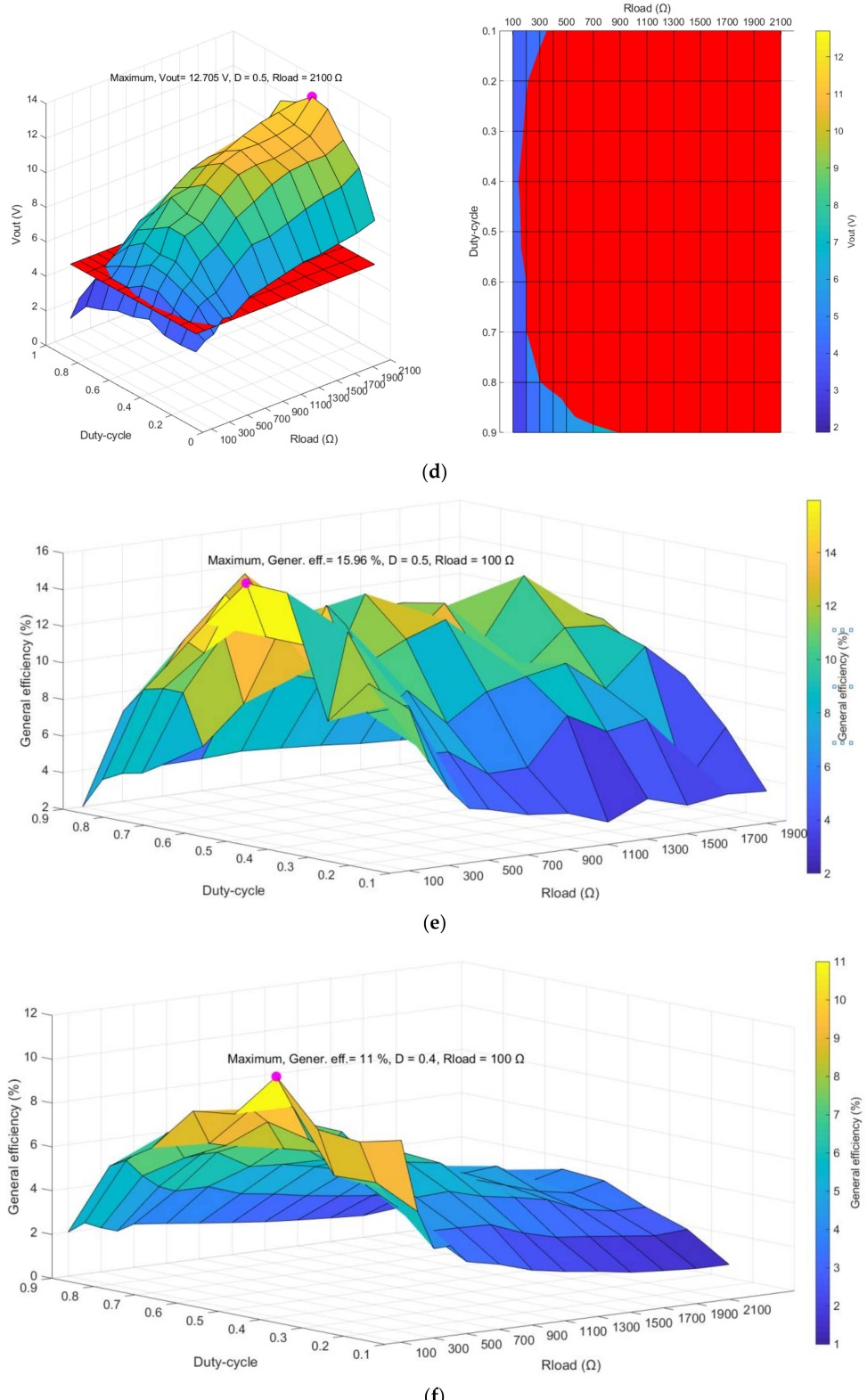

**Figure 34.** Split NP converter simulation results. The (**a**,**c**,**e**) figures are with 3 m/s wind velocity and 200 kHz switching frequency. The (**b**,**d**,**f**) figures are with 10 m/s wind velocity and 500 kHz switching frequency. (**a**) Output power (3 m/s, 200 kHz). (**b**) Output power (10 m/s; 500 kHz). (**c**) Output voltage (3 m/s, 200 kHz), Red plane: 3 V as minimum. (**d**) Output voltage (10 m/s, 500 kHz), Red plane: 3 V as minimum. (**e**) Generator + Converter efficiency (3 m/s, 200 kHz). (**f**) Generator + Converter efficiency (10 m/s; 500 kHz).

### 4.4.4. Summary of Results and Circuit Selection

The results of the converter simulations achieved are summarized in Tables 23 and 24 and Figure 35 that shows the most significant results for the tested maximum and minimum wind velocities. The efficiency of the harvester is set to 100% for all the test cases.

**Table 23.** Converters' best results with the minimum input value: 3 m/s.

| Architec. | $V_{in}$ ($V_{pp}$) | $f_{in}$ (kHz) | $R_{load}$ ($\Omega$) | $D$ | $f_{sw}$ (kHz) | $V_{out}$ (V) | $I_{out}$ (mA) | $P_{out}$ (mW) | $\eta_{generator}$ (%) | $\eta_{converter}$ (%) | $\eta_{system}$ (%) |
|---|---|---|---|---|---|---|---|---|---|---|---|
| Harvester | 0.54 | 1.89 | 27 | - | - | 0.54 | 28.24 | 15.25 | 24.37 | 100 | 24.37 |
| D-bridge | 1.05 | 1.79 | 100 | - | - | 0.57 | 5.71 | 3.26 | 12.35 | 35.02 | 4.32 |
| Side diode | 0.4 | 2.4 | 1500 | 0.5 | 200 | 3.08 | 2.05 | 6.32 | 13.6 | 74.32 | 10.11 |
| Split N | 0.85 | 1.57 | 1300 | 0.5 | 333 | 3.20 | 2.46 | 7.88 | 15.37 | 75 | 11.53 |
| Split NP | 0.68 | 2.12 | 1500 | 0.5 | 200 | 3.41 | 2.62 | 8.95 | 17.61 | 81.52 | 14.36 |

**Table 24.** Converters' best results with the maximum input value: 10 m/s.

| Architec. | $V_{in}$ ($V_{pp}$) | $f_{in}$ (kHz) | $R_{load}$ ($\Omega$) | $D$ | $f_{sw}$ (kHz) | $V_{out}$ (V) | $I_{out}$ (mA) | $P_{out}$ (mW) | $\eta_{generator}$ (%) | $\eta_{converter}$ (%) | $\eta_{system}$ (%) |
|---|---|---|---|---|---|---|---|---|---|---|---|
| Harvester | 2.44 | 8.76 | 27 | - | - | 2.44 | 136.4 | 332.9 | 19.55 | 100 | 19.55 |
| D-bridge | 3.63 | 6.75 | 100 | - | - | 3.1 | 30.6 | 93.7 | 9.72 | 42.64 | 4.15 |
| Side diode | 3.91 | 5.28 | 100 | 0.4 | 500 | 4.19 | 41.95 | 176.0 | 13.01 | 80.04 | 10.46 |
| Split N | 4.09 | 6.42 | 100 | 0.4 | 500 | 4.26 | 42.63 | 181.7 | 13.97 | 76.20 | 10.65 |
| Split NP | 2.31 | 7.77 | 100 | 0.4 | 500 | 4.33 | 43.34 | 187.8 | 14.8 | 74.33 | 11.00 |

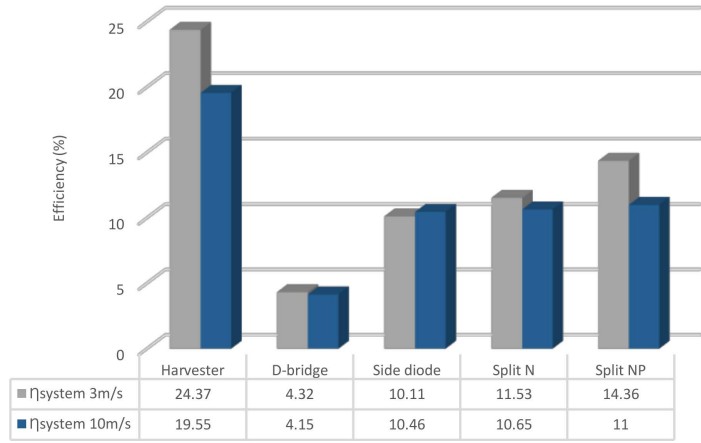

**Figure 35.** System efficiencies with different topologies.

The best system efficiencies (generator + converter) are obtained with Split NP architecture. With 3 m/s wind flow the efficiencies are: 17.61% for the generator, 81.52% for the converter; thus the whole system has an efficiency of 14.36%. With 10 m/s wind flow the efficiencies are: 14.81% of the generator, 74.33% of the converter and 11% of the complete system. In this second case Split NP does not have the best conversion efficiency but it does in the complete system.

Although the D-Bridge was discarded from the beginning, the later statement was verified. The results show that the D-Bridge architecture does not reach the minimum requirement of 3 V at the output point with 3 m/s wind flow; consequently, it would not be valid.

The efficiencies are directly related with the wind flow velocity/pressure condition or input voltage/power level. Furthermore, the same effect occurs with the converters with the selection of MOSFETs. This is because the operating principles of the MOSFET are the maximum responsible for the efficiency of the AC/DC converter.

Based on the results obtained, the Split NP architecture was identified as the AC/DC converter architecture with the best performance, i.e., the highest system efficiency in the considered wind

conditions. Moreover, there are techniques to improve further the efficiency: adaptively set the converter to its maximum efficiency point based on the energy harvester and the load conditions, i.e., an Maximum Power Point Track (MPPT) control and reduce the quiescent consumption.

Finally, the efficiency of the present research is compared with other topologies to verify the improvement achieved. The Table 25 shows this comparison, and the better performance of proposed design.

**Table 25.** Comparison of different converters for Alternate Current/Direct Current (AC/DC) conversion.

| Ref | Topology | Efficiency (%) | Technology | Source |
|---|---|---|---|---|
| [26] | Rectifier + DC/DC (Buck Boost) | 35.64–54.12 | Discrete | Electromagnetic |
| [59] | Rectifier + DC/DC (Frequency up conversion) | 44 | Discrete | Piezoelectric |
| [60] | Rectifier + DC/DC (Buck Boost) | 64.95 | Discrete | Piezoelectric |
| [27] | Rectifier + DC/DC (SECE) | 50–74 | Discrete | Piezoelectric |
| [28] | AC/DC (Boost) | 45 | Discrete | Signal generator |
| This work | AC/DC (Three-phase Split NP) | 74.33–81.52 | Discrete | Electromagnetic |

## 5. Application Examples

One of the application examples, which was taken as reference to develop the work presented in this paper, is a smart pipe of an industrial plant that measures the fluid throughput and thus the plant consumption of such fluid. The wind harvesting system is installed in pipelines, thus, when the gas flow moves the rotor, the harvester generates energy that powers the wireless sensor network. Data will be sent to a computer and/or mobile application, where the fluid quantity will be obtained using mathematical algorithms. Based on the measurement results, a worker or the central system will decide if any action needs to be taken.

Another example of application is the power source of an autonomous system that measures parameters of the bushing of a train, for example, the quality of the wheels. As long as the access of hardwire power wires to this location is complex, it would be desirable to eliminate dependency on batteries, this type of harvester system is an alternative that allows the achievement of the necessary energy to power the sensor system. When the train is moving, an opposite direction air flow is generated, which will fall on the harvester. Hence, the wind harvester rotates, and the AC/DC converter provides the required energy and voltage level to power the system. With node wireless communication, the driver will receive, in real time, the measured data and take actions based on the data collected.

For the least intrusive integration possible, a thin film electronic solution would be needed. The goal would be to integrate the whole sensor, energy harvester, and communications systems into a 100 μm thickness substrate. The flexible electronics would allow the involvement of the wind harvester and easy and small installation in several environments. Figure 36 shows an example of this possible integration.

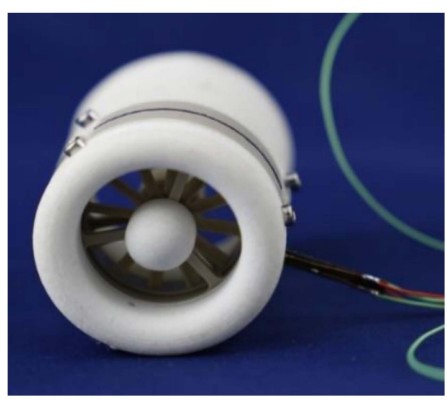 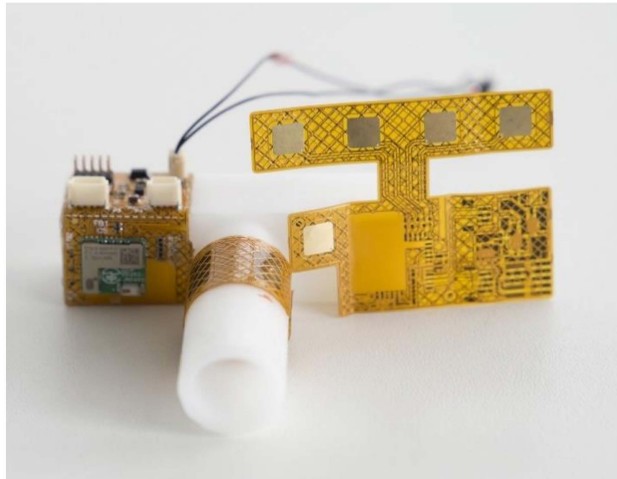

**Figure 36.** Proposal of electronic and wind harvester integration.

## 6. Conclusions

This work proposes and defines a three-phase wind energy harvester system plus different AC/DC converter architectures to collect energy and transform it into DC power.

The AC/DC converter architectures are implemented with different devices, analyzing their main characteristic parameters such as voltage, power and efficiency. Among the architectures considered, the Split NP architecture was chosen as the most suitable to implement the AC/DC converter, for it provides the best performance figures. Test results show that the harvester gives 15–133 mW at two extreme ambient conditions; the aforementioned architecture has an efficiency between 81.52% and 74.33%, better than the standard D-bridge architecture used as reference that provides an efficiency between 35.02% and 42.64%.

The test results also show that the most suitable devices for the AC/DC converter design were: MOSFET Si1553CDLN and diode MBR0520L. The selected devices allow the identification of the key parameters to look at when choosing a MOSFET; these are low on-resistance and low gate capacitance. Likewise, the diode chosen is the one with lower resistance and fastest recovery time. Therefore, the device selection aims to reduce both conduction and switching losses of the converter.

The converter switching frequency must be adapted to the wind conditions obtaining the best results for 200 kHz, 333 kHz and 500 kHz. Moreover, depending on the test case, the most adequate duty-cycle ranges from 0.4 to 0.5. Finally, the loads that provides the best results ranges between 100 $\Omega$ and 1500 $\Omega$.

The proposed energy harvesting system works efficiently at different wind conditions. This research also demonstrates that the AC/DC converters are feasible topologies to be applied in the energy harvesting field and to different types of energy harvesters because the topologies employed in this work were used in the references provided with piezoelectric harvesters that give less energy than a wind harvester. Thus, the proposed system, with a three-phase wind energy harvester system and an AC/DC converter, is a promising technology for multiple small-scale energy applications such as IoT systems and applications to the Industry 4.0 concept.

**Author Contributions:** Conceptualization, B.P., H.Z., and L.M.; methodology, H.Z. and L.M.; validation, B.P., H.Z., J.I.G., and L.M.; formal analysis, B.P., H.Z., L.M., and J.I.G.; investigation, B.P., H.Z., and L.M.; resources, P.S.; data curation, B.P., J.Á.A., and J.I.G.; writing—original draft preparation, B.P.; writing—review and editing, J.Á.A., H.Z., and J.I.G.; supervision, L.M., J.I.G and P.S. All authors have read and agreed to the published version of the manuscript.

**Funding:** This research received no external funding.

**Acknowledgments:** This work was supported by the IK4-TEKNIKER research institute's own funds and the joint work of Electronics and Communications and Intelligent Information System units and by the Department of Education of the Basque Government within the fund for research groups of the Basque university system IT978-16.

**Conflicts of Interest:** On behalf of all authors, the corresponding author states that there is no conflict of interest.

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
