# Peer review of "Mini Wind Harvester and a Low Power Three-Phase AC/DC Converter to Power IoT Devices: Analysis, Simulation, Test and Design"

_applsci, doi:10.3390/app10186347_

Round 1

Reviewer 1 Report

The work presented in this article reports extensively on the analysis of a low power AC/DC converter using different electrical components and simulating for their efficiency alongside a mini wind turbine. The simulation model seems appropriate and the tests conducted are extensive. There are a few areas that the authors can address:

1) discuss the previously published AC/DC converters in Introduction.

2) how does the finalized converter act in comparison to the previously published converters?

3) There are two applications mentioned in Introduction. What are the initial parameters for the wind turbine to harvest energy from these systems? Have you tested the device for this application?

4) How feasible is the fabrication of the circuitry? What will be the size of the final chip?

Author Response

The work presented in this article reports extensively on the analysis of a low power AC/DC converter using different electrical components and simulating for their efficiency alongside a mini wind turbine. The simulation model seems appropriate and the tests conducted are extensive. There are a few areas that the authors can address:

First of all, thanks for your revisions and your suggestions. We have corrected the English style to improve the paper. In the next lines, you have our corrections and answers.

  • discuss the previously published AC/DC converters in Introduction.

The previously published AC/DC converters have been discussed in the reviewed introduction section.

  • how does the finalized converter act in comparison to the previously published converters?

A table has been added at the end of section 4. This table makes the comparison of different converters for AC/DC conversion (some of them have been described in the introduction).

  • There are two applications mentioned in Introduction. What are the initial parameters for the wind turbine to harvest energy from these systems? Have you tested the device for this application?

All the results achieved in this work and presented in the paper are oriented to the refinery application. The application for SHM in the train is only proposed as a potential solution but has not been tested.

  • How feasible is the fabrication of the circuitry? What will be the size of the final chip?

It is not possible to stablish a final size of the circuit due to the IoT system electronics. However, section number 5 has been added, where the two examples are described with an integration solution.

Reviewer 2 Report

This work “Mini wind harvester and a low power three-phase AC/DC converter to power IoT devices: analysis, simulation, test and design. In this manuscript, a three-phase mini wind energy harvester and an AC/DC (Alternate Current/Direct Current) converter for an efficient energy conversion to power different types of electronic systems is presented. This research also analyses in depth a wind harvester operation principle in order to extract its characteristic parameters and propose an equivalent electromechanical circuit of the harvester. Although, the work is interesting, some points should be improved:

  • The abstract of the article can be more concise and must be restructured. Name few of the applications i.e., sensors that can be operated by the designed system for the better understanding of readers at the end of abstract.

  • There are few grammatical errors throughout the paper, which need to be corrected. Try to avoid unnecessary long sentences.

  • Literature review is ambiguous; include some more recent state of the art papers in Literature review for better understanding. Refer following paper for better understanding about wind energy via piezoelectric harvester for IoT:

  • Hassan, Marco Eugeni, and Paolo Gaudenzi. "A review on mechanisms for piezoelectric-based energy harvesters." Energies 11.7 (2018): 1850.

  • The Introduction part of the article must be revised to make it better structured for the readers. Try to explain the previous work related to different aspects of the current research and connect it with the problem statement in the end i.e. identifying the gap and why was this model necessary to develop. An intense revision is required in this section.

  • In the last paragraph of the introduction section, mention the novelty of this paper with previous state of the art research. Rather than mentioning the results/conclusions of the manuscript. Moreover, mention the applications of this work.

  • If possible compare the numerical/experimental data with the data already present in literature for validation in graphical form.

  • Why are you choosing windspeed with 3 and 4 m/s rather than having a wide range of windspeed. Elaborate this with proper citation from the literature.

  • It will be better for readers, if you add optimization graph for resistance, that at which resistance the power will be maximum and after which resistance the power will drop?

Author Response

This work “Mini wind harvester and a low power three-phase AC/DC converter to power IoT devices: analysis, simulation, test and design. In this manuscript, a three-phase mini wind energy harvester and an AC/DC (Alternate Current/Direct Current) converter for an efficient energy conversion to power different types of electronic systems is presented. This research also analyses in depth a wind harvester operation principle in order to extract its characteristic parameters and propose an equivalent electromechanical circuit of the harvester. Although, the work is interesting, some points should be improved:

First of all, thanks for your revisions and your suggestions. In the next lines, you have our corrections and answers.

  • The abstract of the article can be more concise and must be restructured. Name few of the applications i.e., sensors that can be operated by the designed system for the better understanding of readers at the end of abstract.

The abstract has been modified based in the reviewer recommendations.

  • There are few grammatical errors throughout the paper, which need to be corrected. Try to avoid unnecessary long sentences.

We have corrected the English style to improve the paper.

  • Literature review is ambiguous; include some more recent state of the art papers in Literature review for better understanding. Refer following paper for better understanding about wind energy via piezoelectric harvester for IoT: Hassan, Marco Eugeni, and Paolo Gaudenzi. "A review on mechanisms for piezoelectric-based energy harvesters." Energies 11.7 (2018): 1850.

More recent stat of the art paper have been added to the introduction section and the reference about wind piezoelectric harvester for IoT has been also added in the introduction section.

  • The Introduction part of the article must be revised to make it better structured for the readers. Try to explain the previous work related to different aspects of the current research and connect it with the problem statement in the end i.e. identifying the gap and why was this model necessary to develop. An intense revision is required in this section.

The introduction has been remodelled and better structured. Three paragraphs have been added to the introduction such as, description and state of the art of different harvesters, description and state of the art of different converter solutions for this type of harvesters and novelty of the work.

  • In the last paragraph of the introduction section, mention the novelty of this paper with previous state of the art research. Rather than mentioning the results/conclusions of the manuscript. Moreover, mention the applications of this work.

Done, as has been described in the before point.

  • If possible compare the numerical/experimental data with the data already present in literature for validation in graphical form.

A table has been added at the end of section 3. This table makes the comparison of different wind harvesters (some of them have been described in the introduction).

  • Why are you choosing windspeed with 3 and 4 m/s rather than having a wide range of windspeed. Elaborate this with proper citation from the literature.

Extra information has been added to that paragraph, including a new reference:

A quick experiment has been carried out in open space ambient to verify described basics. The experiment has designed with 2 variables: the wind velocity and the resistive load. These 2 variables have been considered due to the results achieved with the theoretical calculus and the minimum range of wind velocity used in other works [38].

  • It will be better for readers, if you add optimization graph for resistance, that at which resistance the power will be maximum and after which resistance the power will drop?

The requested information has been included in the paper at the end part of subsection 3.2.2. The power-resistance relationship is discussed with technical description, 2 figures and 2 tables.

Round 2

Reviewer 2 Report

The reviewer believes that the manuscript has been revised significantly as per the comments. The paper is in good shape now and can be published in the current form. 

However, just a minor point:

References are not according to the journal style. After the 10th reference, the layout is distorted. The authors should correct the reference style and layout as well.